# Topoisomerase 3β interacts with RNAi machinery to promote heterochromatin formation and transcriptional silencing in *Drosophila*

Seung Kyu Lee[1], Yutong Xue[1], Weiping Shen[1], Yongqing Zhang[1], Yuyoung Joo[1], Muzammil Ahmad[1], Madoka Chinen[2], Yi Ding[3], Wai Lim Ku [3], Supriyo De[1], Elin Lehrmann [1], Kevin G. Becker[1], Elissa P. Lei[2], Keji Zhao [3], Sige Zou[4], Alexei Sharov[1] & Weidong Wang[1]

Topoisomerases solve topological problems during DNA metabolism, but whether they participate in RNA metabolism remains unclear. Top3β represents a family of topoisomerases carrying activities for both DNA and RNA. Here we show that in *Drosophila*, Top3β interacts biochemically and genetically with the RNAi-induced silencing complex (RISC) containing AGO2, p68 RNA helicase, and FMRP. *Top3β* and RISC mutants are similarly defective in heterochromatin formation and transcriptional silencing by position-effect variegation assay. Moreover, both *Top3β* and *AGO2* mutants exhibit reduced levels of heterochromatin protein HP1 in heterochromatin. Furthermore, expression of several genes and transposable elements in heterochromatin is increased in the *Top3β* mutant. Notably, *Top3β* mutants defective in either RNA binding or catalytic activity are deficient in promoting HP1 recruitment and silencing of transposable elements. Our data suggest that Top3β may act as an RNA topoisomerase in siRNA-guided heterochromatin formation and transcriptional silencing.

[1] Lab of Genetics and Genomics, National Institute on Aging, National Institutes of Health, Baltimore, MD 21224, USA. [2] Laboratory of Cellular and Developmental Biology, National Institute of Diabetes and Digestive Kidney Diseases, Bethesda, MD 20892, USA. [3] System Biology Center, National Heart, Lung and Blood Institute, National Institutes of Health, Bethesda, MD 20892, USA. [4] Translational Gerontology Branch, National Institute on Aging, National Institutes of Health, Baltimore, MD 21224, USA. Correspondence and requests for materials should be addressed to W.W. (email: wangw@grc.nia.nih.gov)

Topoisomerases, known as the "magicians of the DNA world"[1], can catalyze strand passage reactions for DNA, leading to relaxation of supercoils generated during replication or transcription, and decatenation of tangled DNA during recombination and chromosome segregation. Topoisomerase inactivation can lead to abnormal development, shortened lifespan, lethality, and human diseases[2–4].

Unlike the well-characterized DNA topoisomerases, RNA topoisomerases have drawn little attention for many years. The first eukaryotic RNA topoisomerase, human Top3β, was discovered only recently[4]. Since then, RNA topoisomerase activity has been observed in Type IA topoisomerases from all three domains of life[4–6]. The prevalence of this activity implies that it can provide growth advantage to its host, so that it is retained through millions of years of evolution[7]. The findings also indicate that many Type IA topoisomerases are dual-activity enzymes, capable of solving topological problems for both DNA and RNA. In human, only one of the two Type IA enzymes, Top3β, possesses dual activities, whereas its paralog, Top3α, contains only DNA activity. Top3β but not Top3α contains a conserved RNA-binding domain, RGG-box; and it strongly depends on this domain to bind mRNAs in cells, catalyze RNA topoisomerase reactions, and promote synapse formation[4,8]. Top3β has been purified in a complex with TDRD3 (Tudor domain-containing 3); and this complex biochemically and genetically interacts with FMRP[3,4], an RNA-binding protein (RBP) that is inappropriately silenced in Fragile X syndrome and known to modulate translation of mRNAs important for neurodevelopment and autism[9]. Interestingly, Top3β gene mutation has been linked to schizophrenia and autism[3,5,8], suggesting that Top3β and FMRP may work together to prevent mental dysfunction. Top3β and FMRP bind common mRNAs, associate with mRNA translation machinery, and regulate gene expression at synapse[3–5,8].

In addition to regulating mRNA translation, FMRP also interacts with RNAi machinery to facilitate both transcriptional and post-transcriptional silencing of genes and transposable elements (TEs) in mammals and Drosophila[10–13]. In Drosophila, FMRP has been purified as a component of an RNAi-induced silencing complex (RISC), containing FMRP, Argonaute 2 (AGO2), p68 RNA helicase (encoded by Rm62), Vig (an RBP), and Tudor-staphylococcal nuclease (Tudor-SN)[12–14]. Mutations in RISC and other components of the RNAi machinery disrupt heterochromatic silencing of genes and TEs[15–21].

Here we report the purification of the Top3β–TDRD3 complex from Drosophila S2 cells, and find that it stably associates with RISC. We demonstrate that Top3β mutants display defective heterochromatin formation and transcriptional silencing, which resemble RISC mutants. Moreover, Top3β genetically interacts with RISC to promote heterochromatic gene silencing and recruitment of heterochromatin protein HP1. Our data reveal a function for a dual-activity topoisomerase in RNA metabolism.

## Results

### Drosophila Top3β–TDRD3 complex associates with FMRP and RISC.
We used two different antibodies against Drosophila TDRD3[5], and immunoprecipitated its complex with Top3β from S2 cell lysates. Both antibodies (TDRD3-A and C) isolated two major polypeptides with about equal molar ratio, which were identified as TDRD3 and Top3β by mass spectrometry (MS) (Fig. 1a) and immunoblotting (Fig. 1b). These results are identical to our previous findings for the human complex, and suggest that the complex is conserved in animals.

To identify proteins that associate with the complex, we analyzed the entire TDRD3 immunoprecipitants by MS. We identified peptides from not only FMRP, but also other RISC components—AGO2, p68, and VIG; as well as other FMRP-associated proteins (Fig. 1a). However, the number of peptides derived from RISC were fewer than from either Top3β or TDRD3; and with the exception of FMRP, RISC components were not detectable in the TDRD3 immunoprecipitate by silver-stained analysis. These data suggest that a minor fraction of Top3β–TDRD3 in cells associates with RISC. No peptides from p68 were detected in the immunoprecipitate of TDRD3-A antibody. This could be due to antibody disruption of their association, because this antibody was raised against the region of TDRD3 that binds p68 (see Figs. 1a and 2a). It should be cautioned that although the number of peptides recovered from MS often reflect the protein abundance, they are not always reliable for quantitative comparisons between different samples. We therefore performed IP-Western, which confirmed the presence of several RISC components in TDRD3 immunoprecipitates (Fig. 1b). These data are consistent with our previous findings that Top3β–TDRD3 associates with FMRP in Drosophila[4], and further suggest that a fraction of Top3β–TDRD3 interacts with the FMRP-containing RISC.

To verify the association, we performed immunoprecipitation (IP) by transfecting Flag-tagged Top3β, a Flag-Top3β mutant deleted of its RGG-box, as well as Flag-TDRD3, into S2 cells; and immunoprecipitated the complex with the Flag antibody. MS detected the peptides from not only Top3β and TDRD3, but also FMRP, AGO2, and p68 (Supplementary Fig. 1A and B). As a control, these RISC proteins were absent in mock-IP using the Flag antibody from S2 cells lacking Flag-Top3β or Flag-TDRD3 (Supplementary Fig. 1B). The results indicate that the association between RISC and Top3β–TDRD3 is specific.

We investigated whether the observed association is mediated by RNA by IP–MS using cell extracts untreated or treated with RNase A. We observed a 10-fold reduction in number of peptides derived from FMRP and p68 in TDRD3 IP from RNase A-treated than -untreated extracts (Supplementary Fig. 1C). Similar reduction in RNase-treated extract was also observed in IP using Flag antibodies (Supplementary Fig. 1B). These results differ from those of human studies, where RNase treatment has little effect on the association between Top3β–TDRD3 and FMRP[4], suggesting that the Top3β complex may have diverged interactions and functions with RNA during evolution. We noted that in IP–MS by Flag-Top3β or Flag-Top3β-ΔRGG mutant, the RNase treatment had no significant effect on the number of peptides derived from AGO2, even though the treatment completely eliminated the number of peptides from FMRP and p68 IP (Supplementary Fig. 1B). In IP–MS by Flag-TDRD3 and the two TDRD3 antibodies, the number of AGO2 peptides was reduced by between 30 and 70%, which is smaller than the reduction of peptides from FMRP and p68 (about 90% or more). The data suggest that AGO2 may have direct protein–protein interactions with Top3β and/or TDRD3 that are not mediated by RNA.

### A fraction of FMRP associates with RISC and Top3β–TDRD3.
We next performed reciprocal IP–MS with an FMRP antibody, and obtained three abundant polypeptides, which were identified as FMRP and its two known interactors, Caprin and Rin (Fig. 1a). MS analyses of the entire immunoprecipitate obtained peptides from RISC (AGO2, p68), Top3β and TDRD3, supporting the notion that these proteins associate (Fig. 1a; table). Because the number of peptides from RISC, TDRD3, and Top3β were fewer than that of FMRP (about 5% or less), only a small fraction of FMRP may associate with RISC and Top3β–TDRD3.

We also transfected epitope-tagged RISC components into S2 cells, and performed reciprocal IP with the antibody for the

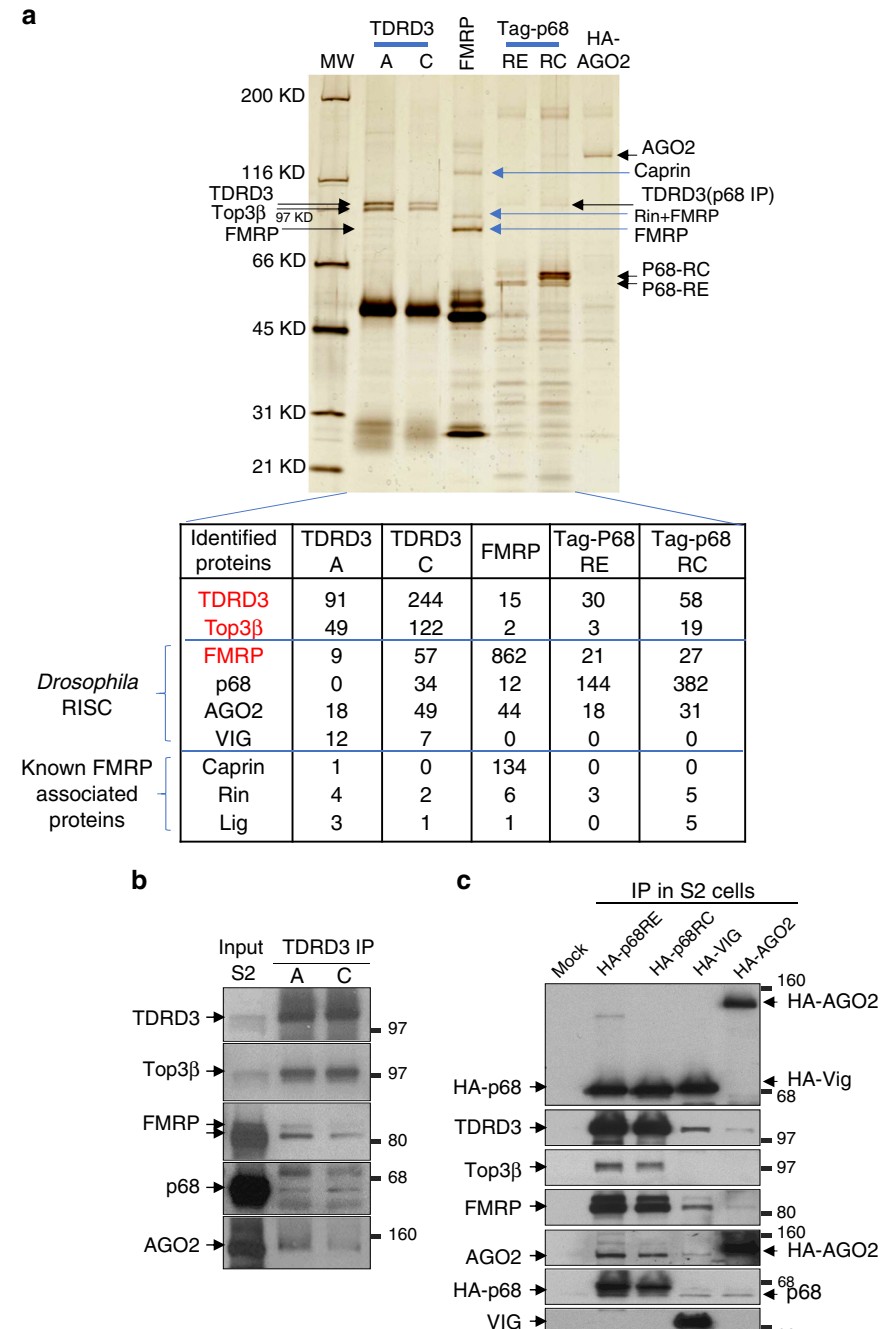

**Fig. 1** Top3β-TDRD3 stably associates with RISC in *Drosophila*. **a** Silver-stained SDS-PAGE analysis of the immunoprecipitated complexes of Top3β-TDRD3, FMRP, p68, and AGO2 from S2 cells; and a partial list of the number of peptides identified by MS analysis of each immunoprecipitate (bottom). The antibodies used for immunoprecipitation are indicated on the top, which include two against TDRD3 (A and C), one for FMRP, and HA antibodies for tagged p68-isoforms (RE and RC) and AGO2. Top3β, TDRD3, components of RISC, and FMRP-associated proteins are indicated by arrows on top and by brackets at the bottom. **b** IP-Western confirms co-IP of Top3β-TDRD3 with different components of RISC by two TDRD3 antibodies. **c** Reciprocal IP-Western shows that Top3β-TDRD3 co-IPs with HA-tagged p68, VIG, and AGO2 using transfected S2 cells and HA antibodies. Note that the level of TDRD3 in each IP is higher than that of Top3β, which is consistent with the MS data

epitope. Silver-staining revealed TDRD3 as a minor polypeptide in the immunoprecipitates of two p68 isoforms (RE and RC) (Fig. 1a). MS identified peptides from RISC, as well as TDRD3 and Top3β (Fig. 1a; table). Immunoblotting further confirmed the presence of these proteins (Fig. 1c). We noted that a previous IP–MS study detected Top3β and TDRD3 in the immunoprecipitate of Flag-AGO2 from S2 cells[22]. In addition, the *Drosophila*

Protein Interaction Map (DPIM) project reported Top3β and TDRD3 peptides in IP by tagged VIG. Consistent with these results, transfection of epitope-tagged-AGO2 and VIG into S2 cells followed by IP-Western obtained TDRD3 (Fig. 1c). Overall, these data support the notion that a fraction of Top3β–TDRD3 associates with RISC. Because the number of TDRD3 peptides is higher than that of Top3β in the immunoprecipitates of FMRP

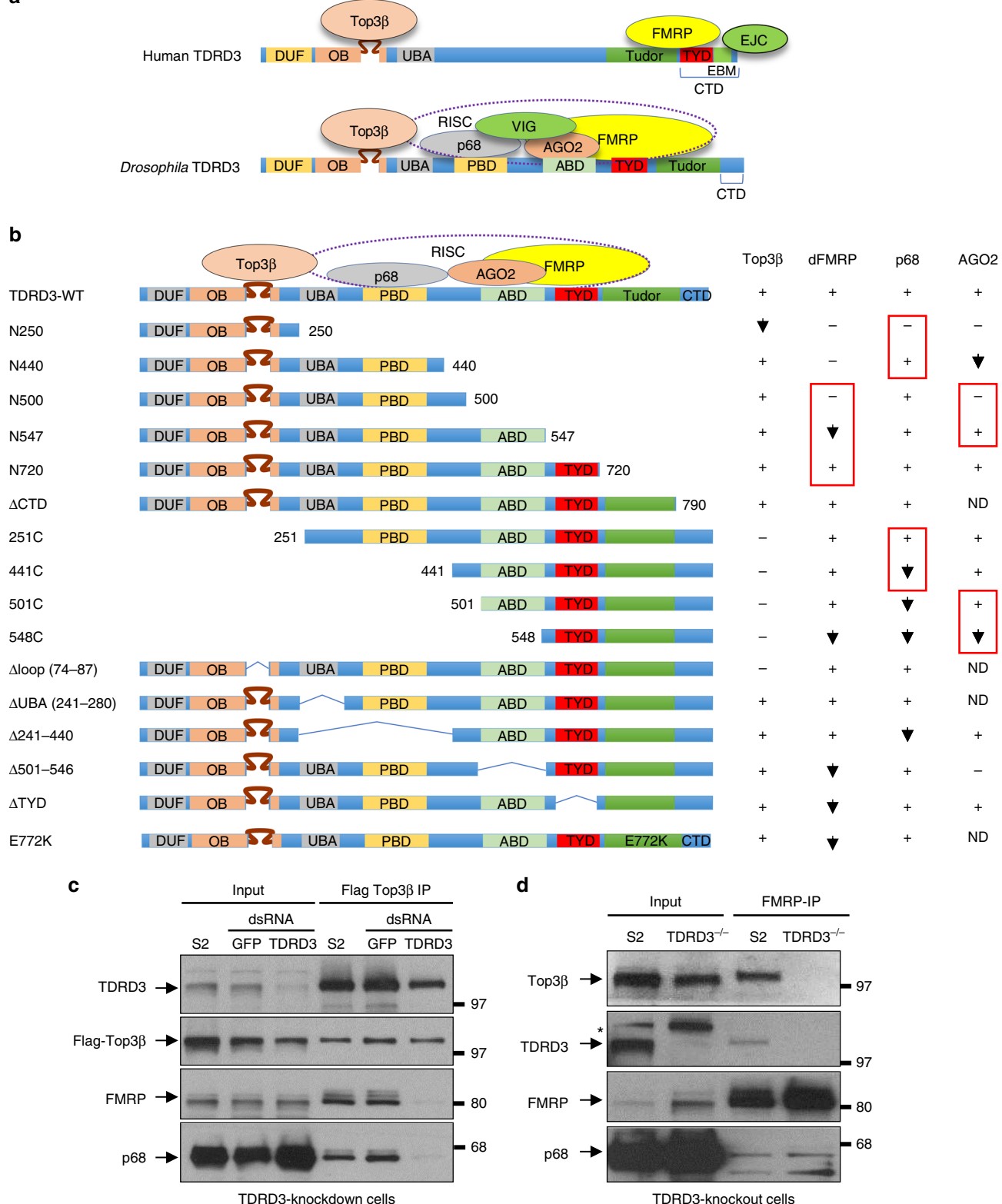

**Fig. 2** TDRD3 acts as a scaffold linking *Drosophila* Top3β to RISC. **a** A cartoon shows similarity and differences between human and *Drosophila* Top3β-TDRD3 complexes (see Results for details). **b** Summary of domain-mapping experiment to show different mutants of TDRD3 (left), and their interactions with Top3β and components of RISC (right). The experiment was based on co-IP between different Flag-TDRD3 constructs and their interacting partners from transfected S2 cells. The detailed IP-Western data are in Supplemental figure (Supplementary Fig. 2A). The presence (+), absence (−), or reduction (down arrow), of interactions are indicated. **c** IP-Western shows that FLAG-Top3β co-IPs with reduced amount of FMRP and p68 in S2 cells depleted with TDRD3 by RNAi. A control experiment was included using RNAi that targets GFP. **d** IP-Western shows that FMRP co-IPs with reduced amount of Top3β, but normal amount of p68, in S2 cells inactivated of TDRD3 by CRISPR

and p68 (about 3–10-fold) (Fig. 1a; table), RISC may have more direct interactions with TDRD3 than Top3β.

**Drosophila TDRD3 acts as a scaffold linking Top3β to RISC.** Human TDRD3 acts as a scaffold, interacting with Top3β and FMRP with its N and C terminal domains, respectively (Fig. 2a)[3,4]. We studied whether *Drosophila* TDRD3 acts similarly by mutating different regions of Flag-TDRD3 (Fig. 2b), transfecting them into S2 cells, and testing their association with Top3β and RISC by co-IP. First, we found that *Drosophila* TDRD3 was identical to its human homolog in requiring the same intervening loop within its OB-fold to associate with Top3β (Fig. 2b; Supplementary Fig. 2A and B; Δloop mutant). Second, *Drosophila* TDRD3 was similar to its human homolog in partial dependence on its Tudor domain for FMRP association (Fig. 2b, Supplementary Fig. S2B; E772K mutant). However, *Drosophila* TDRD3 was independent of its C-terminal domain (CTD) for FMRP association (Fig. 2a, Supplementary Fig. 2B; N790 (ΔCTD)), whereas human TDRD3 was dependent[3,4,23]. Interestingly, the FMRP-interacting motif (FIM) in human TDRD3-CTD was absent in *Drosophila* TDRD3-CTD. Instead, a region at the N-terminus of Tudor, named TYD motif, was found to have sequence similar to human FIM (Fig. 2b, Supplementary Fig. S2C). Deletion of this motif (ΔTYD) reduced the amount of FMRP that co-IPs with Flag-TDRD3 (Supplementary Fig. 2D), suggesting that this motif could be the *Drosophila* FIM. Further mapping revealed that TDRD3 has another region that contributes to FMRP association (between 501 and 547) (Fig. 2b, Supplementary Fig. 2A, E, F; see N500 and N547). These data suggest that TDRD3 in human and flies use different mechanisms to bind FMRP.

We mapped an AGO2-binding domain (ABD) to residues 501–547 of TDRD3 (Fig. 2b; Supplementary Fig. 2A, E, F), the same region that also contributes to FMRP association. This region is well conserved in TDRD3 from different fly species, but poorly in vertebrates (Supplementary Fig. 2G). We also mapped a p68-binding domain (PBD) to residues between 281 and 440 (Fig. 2b; Supplementary Fig. 2A, E, F). This region contains a LFDFL motif, which is conserved among TDRD3 homologs from flies to human (Fig. 2b, Supplementary Fig. 2H), hinting that the association between Top3β–TDRD3 and the RNA helicase may be conserved in animals.

We performed co-IP using *Drosophila* TDRD3-deficient cells[5], and found that Flag-Top3β co-IPed with reduced amount of FMRP and p68 in TDRD3-depleted cells (Fig. 2c). In addition, FMRP co-IPed with reduced amount of Top3β but comparable amount of p68 in TDRD3-knockout cells (Fig. 2d). These data mimics the findings for human TDRD3[3,4], and suggest that *Drosophila* TDRD3 acts as a scaffold linking Top3β to FMRP and p68; but it is dispensable for FMRP–p68 association.

**Top3β and RISC promote heterochromatic gene silencing.** RISC suppresses heterochromatic gene silencing in position effect variegation (PEV) assay[15,19–21,24]. This prompted us to study if Top3β functions similarly using the same assay (Fig. 3a). Briefly, a PEV reporter that encodes *white* gene and localizes near heterochromatin can result in variegated red eyes due to spreading of heterochromatin from regions nearby and subsequent transcriptional silencing in some but not all cells. Mutations in histone methyltransferase (*Su(var)3-9*), RISC, and RNAi biogenesis enzyme *Dcr-2*, can dominantly suppress this phenotype by inhibiting heterochromatin spreading and gene silencing, leading to eyes with more red cells even uniform red eyes. Consistent with earlier reports, introducing a single mutant allele of *AGO2, Rm62* (encoding p68), or *Su(var)3-9*, into the $w^{m4}$ reporter line

suppressed PEV as they produced uniform red eyes (Fig. 3b, top). Importantly, introducing a single copy of any one of the three *Top3β* knockout alleles into this line produced similar uniform red eyes (Fig. 3b, top; Supplementary Fig. 3A), indicating that *Top3β* acts as a dominant suppressor of PEV as does RISC. As a negative control, introducing a wildtype *Top3β* allele in $w^{1118}$ *white* eye mutant background into $w^{m4h}$ line did not alter the eye phenotype (Fig. 3b, Supplementary Fig. 3A; *Top3β* mutants were derived from $w^{1118}$).

We then introduced one *Top3β*-knockout allele into two additional reporter lines in which the *white* gene is located in different pericentric heterochromatin, and observed similar suppression of PEV (Fig. 3b, middle and bottom). We further tested *Top3β* using PEV assays based on two non-*white* reporters, *LacZ* and $Sb^1$, which have a distinct location in heterochromatin compared to the *white* reporters. The advantage of the new reporters is that they can detect modifiers of PEV in tissues and developmental stages different from adult eyes. Notably, a *Top3β* knockout allele increased the reporter expression and thus suppressed PEV in both assays (Fig. 3c and Supplementary Fig. 3B). Together, *Top3β* mutation suppressed five PEV reporters in different heterochromatin regions and in different tissues (Fig. 3d), suggesting that *Top3β* may promote heterochromatic gene silencing throughout *Drosophila* development.

**Top3β has no effect on post-transcriptional silencing.** We examined whether Top3β can act like RISC in RNAi-mediated post-transcriptional silencing by using a reporter line that induces RNAi to post-transcriptionally silence the *white* gene expression, leading to white eye phenotype[25,26]. We found that introducing *Top3β* mutant in the reporter line did not alter the white eye phenotype, whereas introducing the *Dcr-2* mutant produced uniform red eyes (Supplementary Fig. 3C), suggesting that Top3β is dispensable for RNAi-mediated post-transcriptional silencing. The fact that *Top3β*-KO flies have normal post-transcriptional silencing also argue that the observed effect of *Top3β* on heterochromatic gene silencing in PEV assay is not due to defective RNAi biogenesis.

**Top3β and RISC genetically interact in PEV assay.** We next investigated if *Top3β* and RISC genetically interact to suppress PEV by epistasis analysis. We chose the 39C-2 as the reporter, because PEV suppression in this line by *Top3β* mutant generated about 50% of red cells, suitable for detecting either a suppressor or enhancer by the second mutation (Fig. 4a).

First, the introduction of the double heterozygous mutant of $Top3\beta^{-/+};AGO2^{-/+}$ into 39C-2 line enhanced PEV, evidenced by a decrease of the percentage of red cells and appearance of the white eye (Fig. 4b). This result contrasts with those of their single heterozygous mutants, which produce more red cells and thus suppression of PEV (Fig. 4a, b). Extraction of the eye pigment and subsequent measurement confirmed that the red color was decreased in the double but increased in each single mutant (Fig. 4b, right graph). These data suggest that *Top3β* and *AGO2* interact genetically during heterochromatic silencing. We also analyzed interactions between *Top3β* and *AGO1*, which is a homolog of *AGO2* but functions in the miRNA pathway. We found that the $Top3\beta^{-/+};AGO1^{-/+}$ double mutant suppressed PEV similarly as the $AGO1^{-/+}$ single mutant did (Fig. 4c). This result differs from that of the $Top3\beta^{-/+};AGO2^{-/+}$ mutant that enhanced PEV, indicating that the genetic interaction between *Top3β* and *AGO2* is specific.

Second, the introduction of $Top3\beta^{-/+};Rm62^{-/+}$ double heterozygous mutant into the reporter line enhanced PEV (Fig. 4d). This phenotype differs from that of their respective single mutants, but resembles that of $Top3\beta^{-/+};AGO2^{-/+}$ double

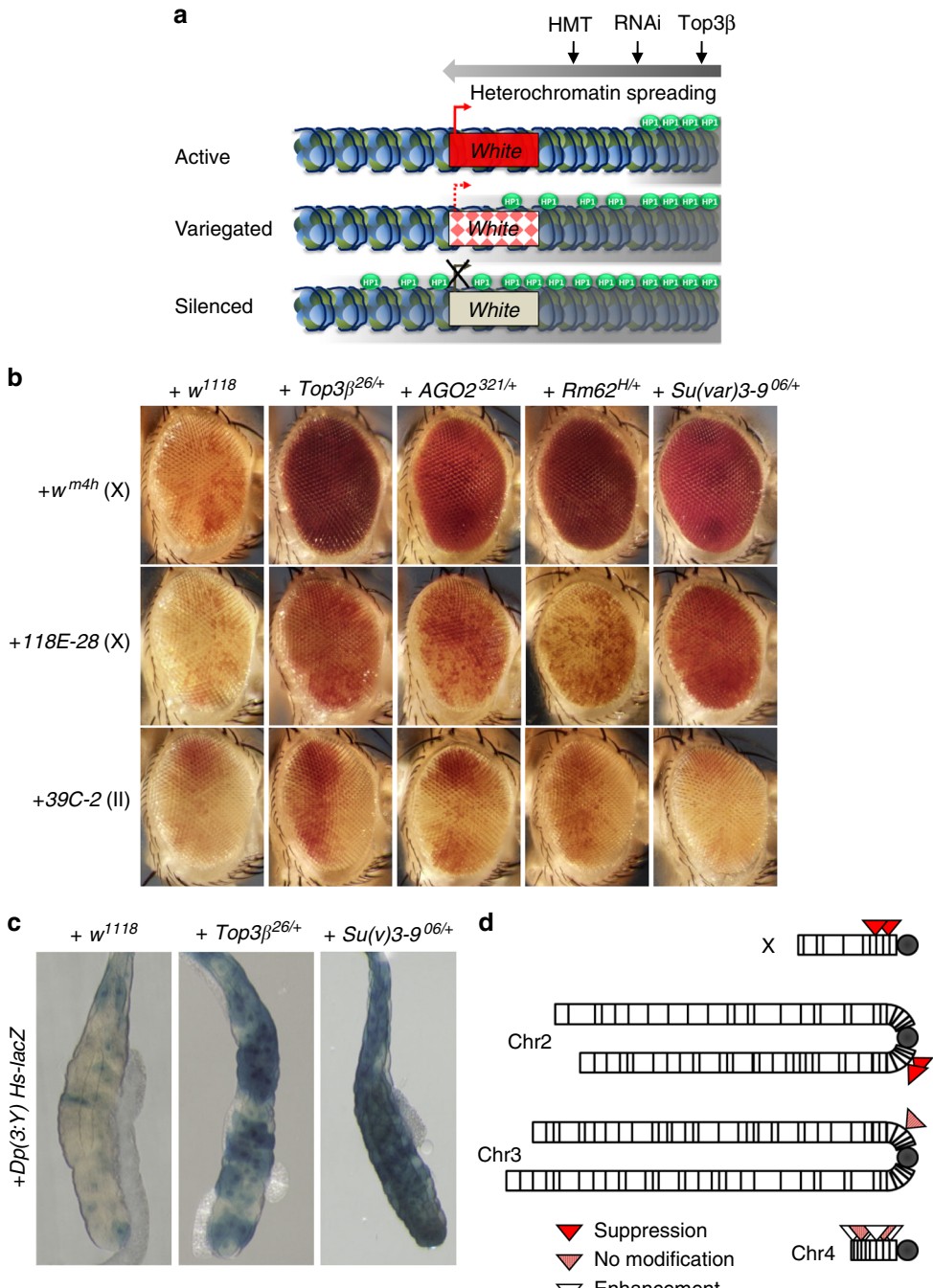

**Fig. 3** *Top3β* and RISC mutants are similar in disrupting heterochromatic gene silencing by PEV assay. **a** A cartoon illustrates the PEV assay using *white* gene as reporters. See Results for description. The heterochromatin is marked by gray area, whereas HP1 is marked by green balls. HMT denotes histone methyltransferase. **b** Representative eye images show that *Top3β* is a suppressor of PEV similarly as are RISC components (*AGO2* and *p68/Rm62*), and *Su(var)3-9*. Increased red color indicates suppression of silencing of *white* gene. Three *white* gene reporter lines were indicated on the left, with letters in parentheses indicating the chromosome with the reporter. Different mutant background is indicated on top. *w1118* serves as a negative control as *Top3β26* was generated in its background. **c** A LacZ reporter-based PEV assay shows that *Top3β* mutant suppresses PEV similarly as does *Su(var)3-9* mutant. PEV suppression is indicated by increased blue color due to de-silencing of the LacZ. *Su(var)3-906/+* served as a positive control, whereas *w1118* as a negative control, as indicated on top. **d** Schematic representation summarizing the PEV modifications by *Top3β* mutants for reporters located on different chromosomes. Note that *Top3β* enhanced PEV for reporters in heterochromatin of chromosome 4, which is known to be different from those on heterochromatin[62]

mutant, suggesting that *Top3β* genetically interacts with both *AGO2* and *Rm62* in heterochromatin silencing.

Third, the introduction of *Top3β−/+;Fmr1−/+* double mutant into 39C-2 line suppressed PEV similarly as their respective single mutants (Fig. 4e), suggesting that *Fmr1* does not have the same genetic interactions with *Top3β* as *AGO2* and *Rm62* do, and that the function of FMRP in RISC is different from that of the other two proteins.

Fourth, the introduction of *Top3β−/+;Dcr-2−/+* double mutant into the reporter line enhanced PEV (Fig. 4f). This phenotype

differs from their respective single mutants, but mimics those of $Top3\beta^{-/+};AGO2^{-/+}$ and $Top3\beta^{-/+};Rm62^{-/+}$ double mutants. These results support a model that siRNAs produced by Dcr-2 form complexes with AGO2, and these siRNA-loaded RISC complexes genetically interact with $Top3\beta$ to promote heterochromatin formation and gene silencing.

**Heterochromatin is abnormal in $Top3\beta$ mutant flies.** RISC mutants have aberrant cellular distribution of heterochromatin markers, including HP1 and H3K9me2[15,20]. We analyzed these markers by immunostaining salivary glands of the $Top3\beta$ mutant. In agreement with previous data[27], both markers displayed concentrated staining in the control flies (Supplementary Fig. 4A, B),

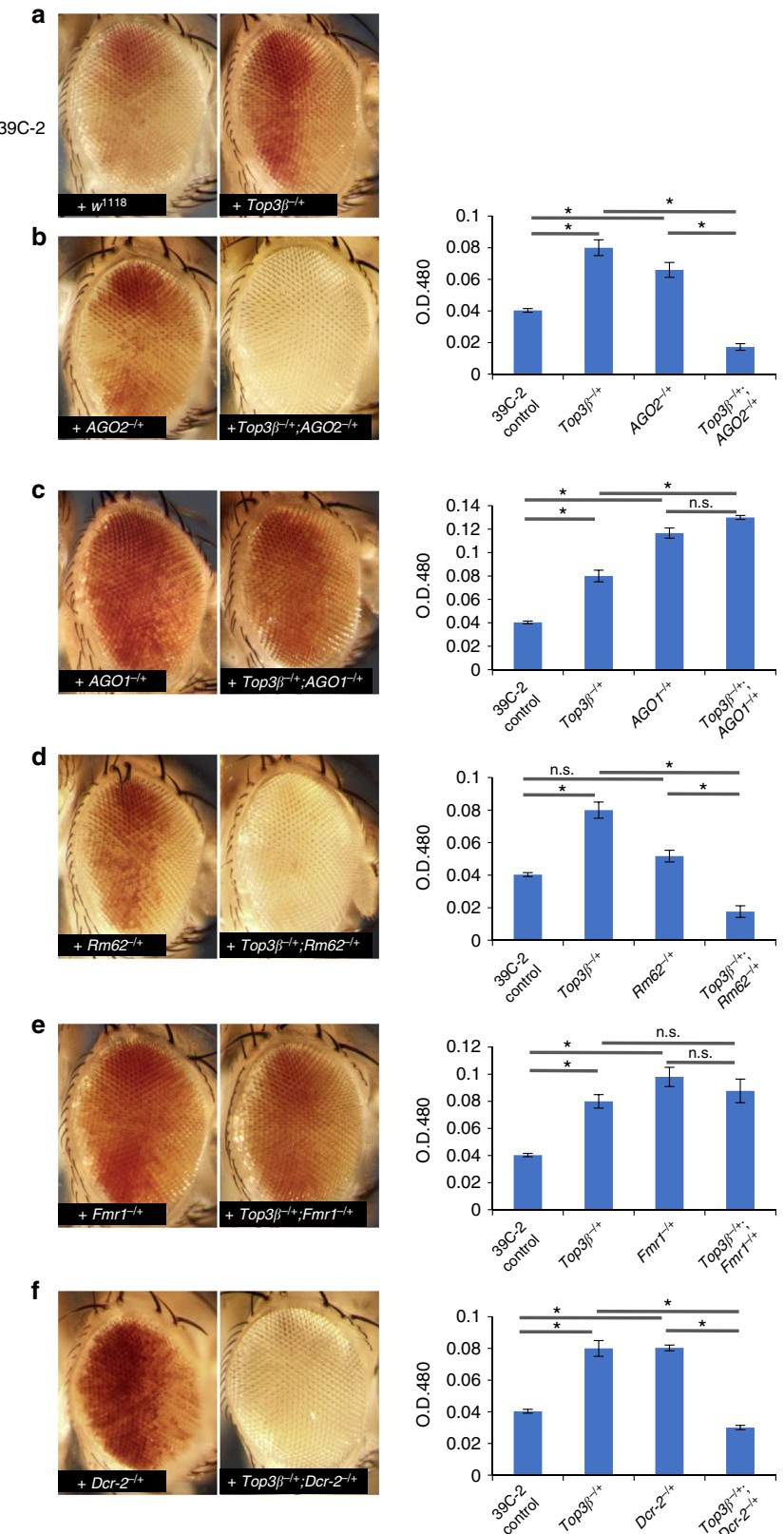

which are due to clustering of pericentric heterochromatin of different chromosomes. Notably, both markers showed less concentrated staining in *Top3β* mutant. As a control, histone H3 showed similar staining in both mutant and the control flies. Quantification confirmed that the ratios between the signals of HP1 or H3K9me2 versus that of H3 were reduced in *Top3β* mutant. Immunoblotting revealed that *Top3β* mutant had normal levels of the two markers (Supplementary Fig. 4C), indicating that *Top3β* mutation does not affect HP1 synthesis or global H3K9 methylation. Together, the results imply that *Top3β* mutation may disrupt HP1 recruitment and H3K9 methylation in pericentric heterochromatin, which is consistent with the PEV data.

**Top3β and RISC promote HP1 recruitment to heterochromatin**. We next performed HP1 ChIP-seq (chromatin immunoprecipitation followed by high-throughput sequencing) to identify the defective heterochromatin in *Top3β* and *AGO2* mutants at the molecular level. Our analyses were focused on chromosomes 2 and 3, because their pericentric heterochromatin is long and distinguishable from euchromatin.

Consistent with earlier reports[28,29], the ChIP signals of HP1 were enriched in pericentric heterochromatin in control and *Top3β* mutant flies, after normalization with their respective input signals (Supplementary Fig. 5A, top). We then calculated the difference in HP1 signals between each mutant and the control ($w^{1118}$) using the SICER (Spatial-clustering method for Identification of CHIP-Enriched Regions) program, and displayed the HP1-decreased islands (HDIs) and HP1-increased islands (HIIs) by blue or red lines, respectively, in bedgraphs. The analysis revealed much more HDIs (blue) than HIIs (red) in pericentric heterochromatin of both *Top3β* and *AGO2* mutants (Fig. 5a; Supplementary Fig. 5B). There was also an HDI at telomeric heterochromatin of chromosome 3L (Fig. 5a). Bedgraphs of sequencing reads in 100-bp windows across two telomeric and one representative pericentric heterochromatin regions revealed that many HP1 peaks present in the control flies are concomitantly reduced in both *Top3β* and *AGO2* mutants (Fig. 5b–d), which is consistent with the SICER data and suggests that Top3β and RISC can work together to recruit HP1 to the same loci. The reduction of HP1 signals was confirmed by ChIP-qPCR for two representative pericentric heterochromatin loci (see Fig. 6d, e).

**More defective HP1 recruitment in AGO2 than Top3β mutant**. We observed that in pericentric heterochromatin, there are more HDIs in *AGO2* than *Top3β* mutant (Fig. 5a, Supplementary Fig. 5B). In addition, the height of HP1 peaks was reduced to a greater extent in the *AGO2* than *Top3β* mutant (Fig. 5a, Supplementary Fig. 5B). These data suggest that the effect of RISC on HP1 recruitment is stronger than that of Top3β.

Quantification further revealed the differential effects of the two mutants on HP1 recruitment. First, the sum of HDIs represent 23.6% and 43.3% of all HP1 islands in *Top3β* and *AGO2* mutants, respectively; which are about 500- and 100-fold higher

than the percentages of HIIs (0.04% and 0.4%, Fig. 5e), respectively; indicating that the predominant effect of Top3β and RISC on heterochromatin is to increase HP1 recruitment. However, over 55% of HP1 islands showed no change in either mutant, arguing that Top3β and RISC are necessary for recruiting HP1 to some but not all loci. Second, the sum of HDIs accounted for 13.6% and 37.0% of the entire length of pericentric heterochromatin in *Top3β* and *AGO2* mutants, respectively (Fig. 5f), consistent with the SICER bedgraphs showing more HDIs in *AGO2* than *Top3β* mutant (Fig. 5a, Supplementary Fig. 5B). Of all the HDIs, 3.6% was present in *Top3β* mutant only, 27% in *AGO2* mutant only, whereas 10% were present in both (Fig. 5f). The HDIs present in both mutants represented 74% and 27% of the total HDIs in *Top3β* and *AGO2* mutants, respectively. The data suggest that a large fraction of Top3β and a small fraction of RISC may work together to recruit HP1 to the same loci, whereas a small fraction of Top3β and majority of RISC can act independently to recruit HP1.

**Top3β and AGO2 genetically interact for HP1 recruitment**. To study whether *Top3β* and *AGO2* genetically interact in pericentric heterochromatin formation as suggested by PEV data, we analyzed their double mutant by HP1 ChIP-seq. The number of HDIs (blue) in pericentric heterochromatin was fewer in the double mutant than those of their single mutants (Fig. 5a, Supplementary Fig. 5B). In addition, many HP1 peaks reduced in each single mutant were not reduced in the double mutant (Fig. 5d, Supplementary Fig. 5C). Quantification of SICER data showed that while HDIs in each single mutant accounted for >20% of total HP1 islands, the corresponding percentage in the double mutant was 1.2%, a decrease of 20-fold or more (Fig. 5e). Concomitantly, the percentage of HP1 islands of no change was increased in the double mutant (98.6%) when compared to their single mutants (76.4% and 56.3%). Moreover, the ratio between HDIs and HIIs is about 6 for the double mutant, which is considerably smaller than those of *Top3β* and *AGO2* single mutants (about 500 and 100, respectively) (Fig. 5e). These data suggest that the defective heterochromatin phenotype is largely suppressed in the double mutant. The data are reminiscent of those of the PEV assay (Fig. 4a, b), suggesting that Top3β and RISC genetically interact to promote HP1 recruitment to pericentric heterochromatin.

**Top3β, AGO2, and HP1 binding sites overlap in heterochromatin**. If Top3β and RISC directly promote HP1 recruitment, their binding sites on chromatin may overlap with those of HP1. To test this hypothesis, we identified genome-wide Top3β and AGO2 binding sites in fly heads by ChIP-seq. For AGO2 ChIP, we utilized a fly line that has a Flag-tag knocked-in and fused in-frame to the endogenous *AGO2*[30]. SICER analysis showed that AGO2 and Top3β ChIP islands are present in heterochromatin at much lower frequency and scores than those of HP1 (Supplemental Fig. 6A; HP1 islands used a scale 100-fold higher than the others). Specifically, the percentages of AGO2 and Top3β ChIP islands in total heterochromatin were about

**Fig. 4** *Top3β* and RISC genetically interact to promote heterochromatin gene silencing by PEV assay. **a** Representative eye images show suppression of PEV reporter 39C-2 by heterozygous mutant of *Top3β*[26]. **b** Representative eye images (left), and pigment quantification assay (right), show that double heterozygous mutants of *Top3β*[26/+]; *AGO2*[321/+] enhanced PEV (white eyes) for the 39C-2 reporter line. **c** Same as **b**, except that mutants of *Top3β*[26] and *AGO1*[04845] were analyzed. **d** Same as **a**, except that *Top3β* and *Rm62* mutants were analyzed. **e** Same as **b**, except *Top3β* and *Fmr1* mutants were analyzed. **f** Same as **b**, except that *Top3β* and *Dcr-2* mutants were analyzed. Several images of control and single heterozygous mutants were copied from Fig. 3b for the convenience of readers. The asterisks in graphs mark *p*-values less than 0.05 by Student *T*-test. n.s. represents statistically non-significant *p* values ($p > 0.05$). The error bars are obtained by standard error method. The results are obtained from three independent experiments

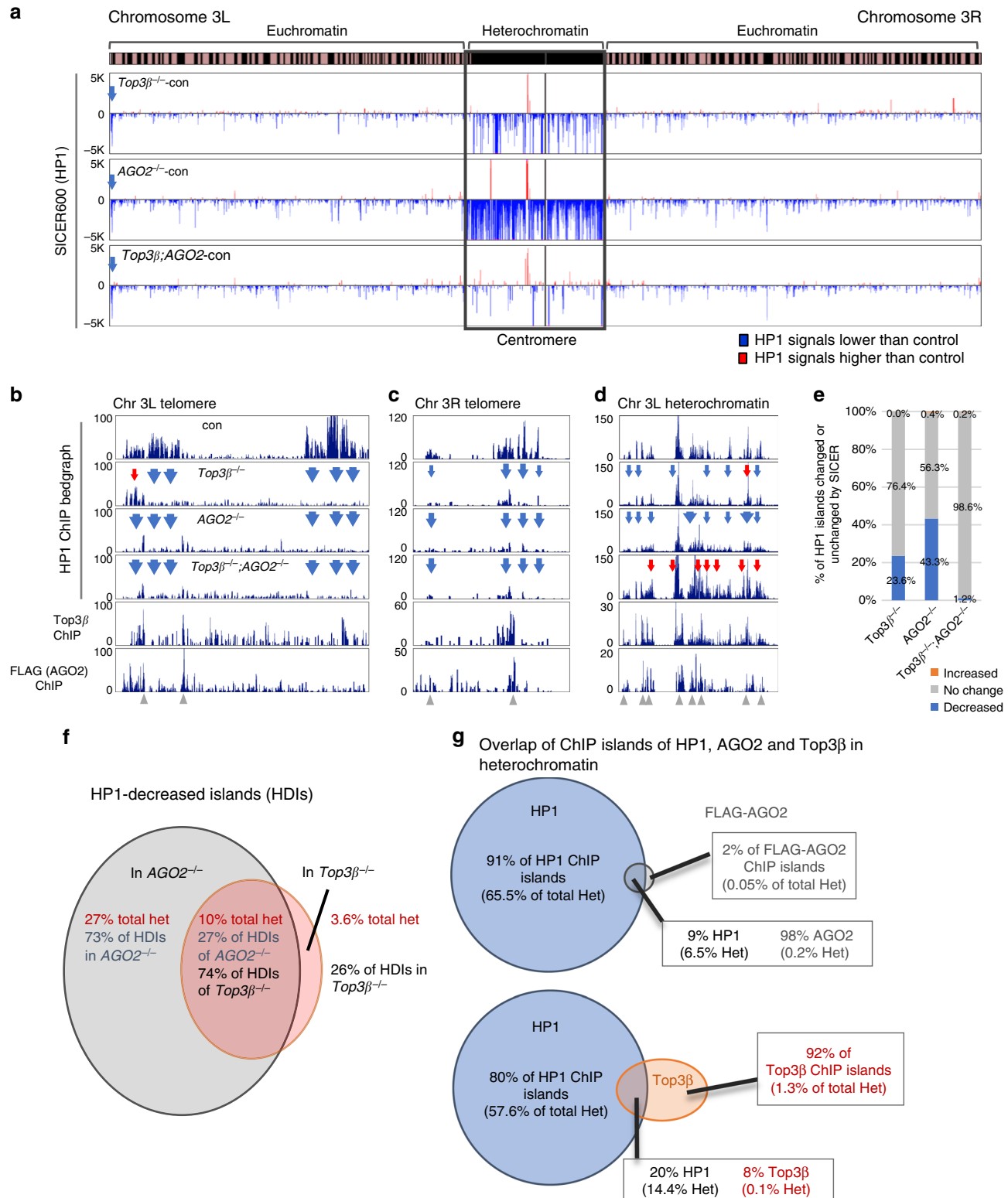

0.3% and 1.4%, respectively (Fig. 5g), which are substantially smaller than that of HP1 (72%), as well as that of HDIs in total heterochromatin of *AGO2* and *Top3β* mutants (37% and 13.6%, respectively; Fig. 5f). The data are consistent with a mechanism of "nucleation and spreading[31]": RISC and Top3β may bind a small number of loci to nucleate the initial assembly of hetero-chromatin, which may then recruit additional components to spread to other regions.

Interestingly, almost all AGO2 islands (98%) in pericentric heterochromatin overlap with those of HP1 (Fig. 5g), supporting a role of chromatin-bound RISC in HP1 recruitment. In contrast, a much smaller percentage of Top3β islands (8%) overlap with those of HP1 (Fig. 5g) or AGO2 (Supplemental Fig. 6B), arguing that a minor fraction of chromatin-bound Top3β may be involved in the same process. Inspection of selected regions in heterochromatin revealed that some HP1-reduced peaks in *Top3β*

**Fig. 5** Top3β works with AGO2 to facilitate HP1 recruitment to specific loci in heterochromatin. **a** Bedgraphs displaying HP1 SICER score difference (*Y*-axis) on chromosome 3 between the single or double mutants of *Top3β* and *AGO2* versus the control (*w1118*). The *X*-axis indicates chromosome location of the HP1 islands on dm6 version of *Drosophila* genome. The HP1-decreased and increased islands (HDIs and HIIs) are marked by blue and red, respectively. An arrow indicates an HDI in 3L telomeric region. The chromosome bands indicate pericentric heterochromatin (black) from euchromatin (striped). **b–d** The bedgraphs of HP1 ChIP-seq reads in FPKM (Fragments Per Kilobase of genome per Million mapped reads) in representative regions of chromosome 3L telomere (**b**), 3R telomere (**c**), and 3L pericentromeric region (**d**) for the single and double mutants of *Top3β* and *AGO2*, and the control. Blue and red arrows indicate reduced and increased HP1 signals in the mutants, respectively. Window width is 100 bp for the calculation. The bottom two panels of **b–d** show Top3β and FLAG-AGO2 ChIP-seq signals in the same locus. Note that anti-FLAG antibodies were used for ChIP-seq experiment in a fly line with a Flag-tag knocked-in at the AGO2 gene. The green arrows indicate the overlap between AGO2, Top3β, and reduced HP1 peaks. The error bars represent standard errors. **e** A graph shows the percentage of HP1 islands that are decreased (blue), increased (red), or no change (gray) of chromosomes 2 and 3 for each mutant compared to the control. The quantification is based on average of SICER scores from 3 independent ChIP experiments for *Top3β*, 2 for *AGO2*, and 2 for *Top3β/AGO2* mutants. **f** A Venn diagram shows mean percentages of HDIs in total pericentric heterochromatin of chromosomes 2 and 3 for single *Top3β* or *AGO2* mutant only. The quantification is based on SICER scores. **g** Venn diagrams show the overlap of ChIP islands and the total length of these islands between HP1 and AGO2 (top), and Top3β (bottom) in pericentric heterochromatin of chromosomes 2 and 3 of the *w1118* line. The quantification is based on the length from the SICER analysis

and *AGO2* mutants overlap with AGO2 and Top3β binding sites (Fig. 5b–d), supporting the notion that RISC and Top3β work together to promote HP1 recruitment.

**Top3β and AGO2 recruit HP1 to some loci in euchromatin**. Our HP1 ChIP assay detected a small number of HDIs and HIIs in euchromatin of *Top3β* and *AGO2* mutants (Fig. 5a and Supplementary Fig. 5B). Notably, the percentage of HDIs (1.3–2.1%) was about 40–50-fold higher than HIIs (0.03–0.04%) (Supplementary Fig. 5D), suggesting that Top3β and RISC can also promote HP1 recruitment to euchromatin, albeit at fewer loci than heterochromatin. Some HDIs were concomitantly present in single and double mutants of *Top3β* and *AGO2* (Fig. 5a and Supplementary Fig. 5B–C), suggesting that Top3β and RISC may work in the same pathway in HP1 recruitment at these loci. In agreement with this notion, a fraction of AGO2 and Top3β ChIP islands were found to overlap with those of HP1 and each other in euchromatin (Supplementary Fig. 6C and D); and inspection of representative euchromatin regions revealed that some AGO2 and Top3β binding sites overlap with HP1-reduced peaks in *Top3β* and *AGO2* mutants (Supplementary Fig. 5C).

We noticed that majority of Top3β and AGO2 ChIP islands are localized in euchromatin (Supplementary Fig. 6A). This is in contrast to HP1 islands, most of which are in heterochromatin, implying that majority of Top3β and RISC have functions in euchromatin that are unrelated to HP1 recruitment. Indeed, only a minor fraction (16%) of AGO2 islands overlap with those of HP1 in euchromatin (Supplementary Fig. 6C), which is much smaller than the 98% observed in heterochromatin, consistent with previous findings that AGO2 can function independent of HP1 recruitment[32,33].

**Top3β promotes H3K9 methylation**. HP1 is known to bind H3K9me3 in heterochromatin. Our data that H3K9me2 immunostaining is reduced in the *Top3β* mutant prompted us to investigate whether H3K9 methylation is decreased in pericentric heterochromatin by ChIP-seq. We observed enrichment of H3K9me2 and H3K9me3 in pericentric heterochromatin, which is similar to that of HP1 (Supplementary Fig. 5A). SICER analysis revealed more decreased (blue) than increased (red) islands for both methylation marks in pericentric heterochromatin of the *Top3β* mutant (Fig. 6a and Supplementary Fig. 7), indicating that Top3β is needed for normal H3K9 methylation in heterochromatin. The percentages of islands with decreased H3K9me2 and H3K9me3 signals were 7.0% and 4.6%, respectively, among all their respective islands (Fig. 6b), which are about 18- and 23-fold higher than percentages of the islands with increased signals

(0.4% and 0.2%, respectively; Fig. 6b, c). These differences are smaller than that of HP1 (~500-fold) (Fig. 5e), suggesting that Top3β is more important for HP1 recruitment than H3K9 methylation.

We then selected two representative loci from pericentric heterochromatin of 3L and 2R based on the presence of HDIs (Fig. 6d, e; left), and found that both loci exhibited concomitant reduction of H3K9me2 and H3K9me3 signals by bedgraph analyses and ChIP-qPCR (Fig. 6d, e; right). Together, these data suggest that Top3β is needed for HP1 recruitment and H3K9 methylation at the same loci in heterochromatin.

We observed smaller changes in H3K9me2 and H3K9me3 islands in euchromatin when compared to those in heterochromatin (Fig. 6a, b; Supplementary Fig. 7). The ratios between the percentages of decreased and increased islands are 0.7 and 1.1 (Fig. 6c), respectively, which are considerably smaller than the 18- and 23-fold ratios for the two marks in heterochromatin (Fig. 6b). These data suggest that the major effect of Top3β on HP1 recruitment and H3K9 methylation is in heterochromatin.

**Top3β mutant displays defective transcriptional silencing**. As defective heterochromatin formation often leads to defective silencing of genes and TEs[34,35], we investigated whether *Top3β* mutant has this defect at several heterochromatin loci where HP1 level is reduced. In the telomeric region of chromosome 3L, three neighboring genes (*Lsp1γ*, CG13405, and *mthl8*) were found to be upregulated by RNA-seq, microarray assays (Fig. 7a), and RT-qPCR (Fig. 7b). In addition, seven TEs located between *mthl8* and *Lsp1γ* were also found to be upregulated in the *Top3β* mutant (Fig. 7c). As a control, a gene located outside of this locus (*pk61c*) showed no obvious change. Similarly, in the telomeric region of chromosome X, two neighboring genes were upregulated (Supplementary Fig. 8A, 8B). Moreover, RNA-seq and RT-qPCR revealed that two TEs located in pericentric heterochromatin, *Doc* and *gypsy1*, are consistently de-silenced in *Top3β* mutant flies (Fig. 7d, e). Together, these data support our PEV data that Top3β is needed for transcriptional silencing.

**Top3β function requires RNA binding and catalytic activity**. We have previously shown that Top3β depends on both its RNA binding and catalytic activities to promote synapse formation using transgenic rescue experiment[8]. Here we used the same strategy to study whether Top3β depends on the same activities to mediate heterochromatin formation and transcriptional silencing. Briefly, we transgenically expressed in the *Top3β* mutant background the wildtype and two mutant versions of Top3β-a catalytic point mutant (Y332F) that contains normal mRNA binding

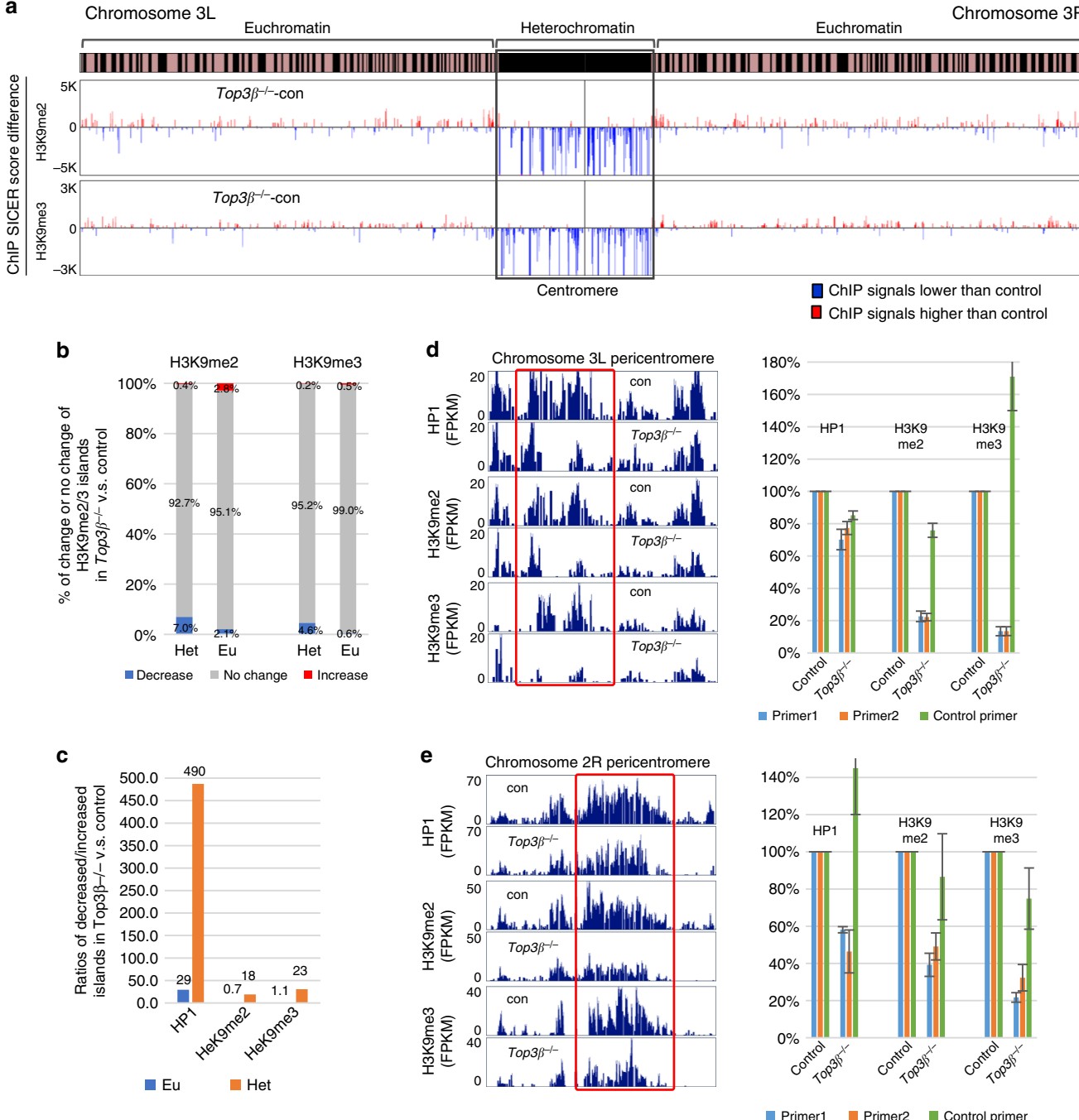

**Fig. 6** Top3β promotes H3K9 methylation and HP1 recruitment in heterochromatin. **a** Bedgraphs show differences of SICER scores of H3K9me2 and H3K9me3 ChIP-seq data between *Top3β* mutant versus the control (*w[1118]*) for chromosome 3 (Y-axis). Islands with decreased and increased signals are marked by blue and red, respectively. **b** A graph shows the percentage of H3K9me2 or H3K9me3 islands that are decreased (blue), increased (red), or of no change (gray) in heterochromatin (Het) or euchromatin (Eu) of *Top3β* mutant when compared to the control. The quantification is based on SICER analysis from 3 independent ChIP experiments. **c** A graph shows the ratio of islands with decreased over those with increased signals of HP1, H3K9me2, and H3K9me3 in heterochromatin (red) and euchromatin (blue) of *Top3β* mutant when compared to the control. The quantification for HP1 is based on 3 ChIP-seq experiments. **d**, **e** Bedgraphs of ChIP-seq data (left) and graphs of RT-qPCR (right) show that two representative pericentric heterochromatin regions on chromosome 3L and 2R have decreased signals of HP1, H3K9me2, and H3K9me3 in *Top3β* mutant when compared to the control. The red box marks the region in which 2 primer pairs for qRT-PCR were selected. A pair of control primers was selected from the euchromatin of the corresponding chromosomes. The results of RT-qPCRs were from triplicates. The error bars are obtained by standard error method

activity but lacks topoisomerase activity for both DNA and mRNA, and an RGG box-deletion mutant (ΔRGG) that has strongly reduced mRNA binding, as well as decreased topoisomerase activity for both nucleic acids (Fig. 8a, b)[4,8]. ChIP assays revealed that the reduced HP1 levels at the two representative loci

in pericentric heterochromatin in the *Top3β* mutant (Fig. 6d, e) were rescued by the wildtype but not two mutants of Top3β (Fig. 8c, d). In addition, *Top3β* wildtype transgene increased HP1 signals at a telomeric locus of X chromosome to a level that is higher than that of the control line, whereas its two mutants

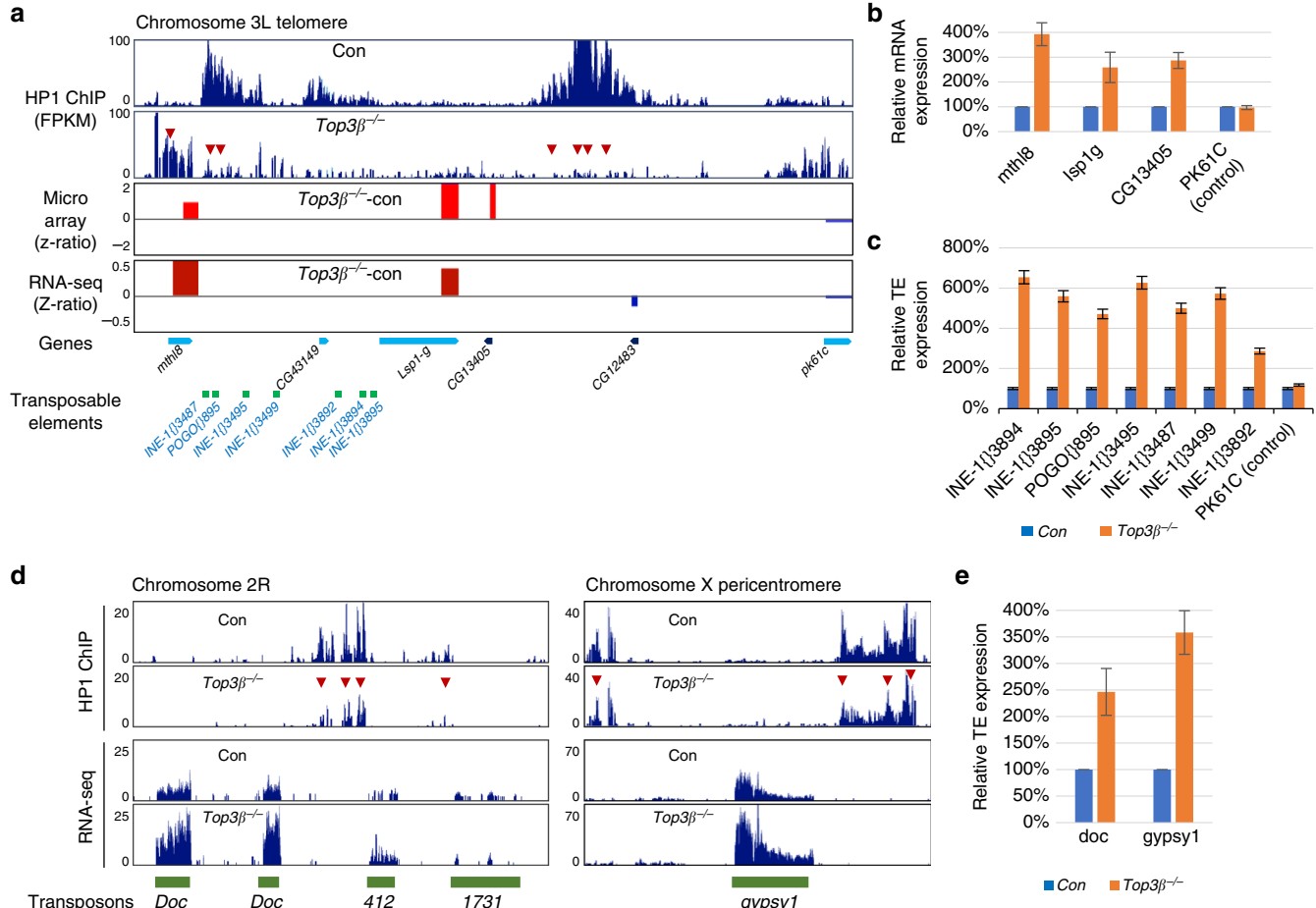

**Fig. 7** Top3β is required for heterochromatin formation and gene silencing at telomeres. **a** Bedgraphs of ChIP-seq, RNA-seq, and cDNA microarray data show reduced HP1 signals and increased gene expression in the telomeric region of chromosome 3L. Arrows in the ChIP-seq bedgraph indicate the regions with reduced HP1 signals in Top3β mutant when compared to the control ($w^{1118}$). The bedgraphs of RNA-seq and cDNA microarray used z-ratios (y-axis) to show the difference in transcript levels between the Top3β mutant and the control for several neighboring genes in this region. The increased and decreased RNA signals in Top3β mutant are marked by red (increase) and blue (decrease), respectively. The chromosome location of genes and TEs are indicated at the bottom graph. **b**, **c** Graphs of RT-qPCR data show de-silencing of the genes (**b**) and TEs (**c**) in chromosome 3L telomere as listed in **a**. A negative control gene (pk61c) was included, which is located outside of the HP1-decreased region. **d** Bedgraphs of ChIP-seq and RNA-seq data show reduced HP1 signals and increased the levels of TEs in chromosome 2R (left) and chromosome X (right) pericentric heterochromatin regions. The red arrows indicate decreased HP1 ChIP signals in the Top3β mutant. **e** Graphs of RT-qPCR data show de-silencing of TEs expression. RT-qPCR was performed in triplicates and TE levels were measured by normalizing with rp49. The error bars are obtained by standard error method

were defective (Fig. 8e). These data suggest that Top3β depends on both RNA-binding and catalytic activities to promote HP1 recruitment to specific loci in heterochromatin.

We also found that the two TEs de-silenced in the Top3β mutant (Doc and gypsy1, Fig. 7e) were largely repressed by transgenic expression of the wildtype Top3β (Fig. 8f). In contrast, this repression was reduced by transgenic expression of the two mutant proteins (Fig. 8f). These data suggest that Top3β depends on its RNA-binding and catalytic activities for transcriptional silencing.

## Discussion

Recent evidence suggests that Top3β has two functions: one for DNA, where it reduces negative supercoiling to resolve R-loops[36]; and the other for mRNAs, where it associates with polyribosomes and FMRP to regulate translation[3–5]. Here we show that Top3β has an additional function—it interacts with RISC to facilitate heterochromatin formation and transcriptional silencing. Because Top3β is a dual-activity enzyme, one key question is whether it acts on DNA or RNA during this process. Our findings that

Top3β mutant deleted of its RNA-binding domain (RGG-box) is deficient in promoting HP1 recruitment to heterochromatin and in silencing of TEs suggest that Top3β acts on RNA. This notion is further supported by the data that Top3β biochemically and/or genetically interacts with an RNA slicer (AGO2), an RNA helicase (p68), and an RNase (Dcr-2). However, because the RGG-box deletion mutant not only has defective RNA-binding activity, but also reduced catalytic activity for both DNA and RNA, it is possible that the reduced catalytic activity on either DNA, or RNA, or both, is responsible for the defective heterochromatin formation. A mutation that inactivates the catalytic activity of Top3β on one nucleic acid but not the other is needed to clarify this issue.

Another question is whether Top3β acts as a topoisomerase or an RNA-binding protein. Our findings that Top3β-Y332F mutant, which has normal RNA binding but no catalytic activity, is deficient in promoting HP1 recruitment and TE silencing suggest that Top3β acts as a topoisomerase. The next question is where the topological problem may come from? The current hypothesis on heterochromatin formation postulates that siRNA-loaded RISC may guide histone methyltransferases and HP1 to

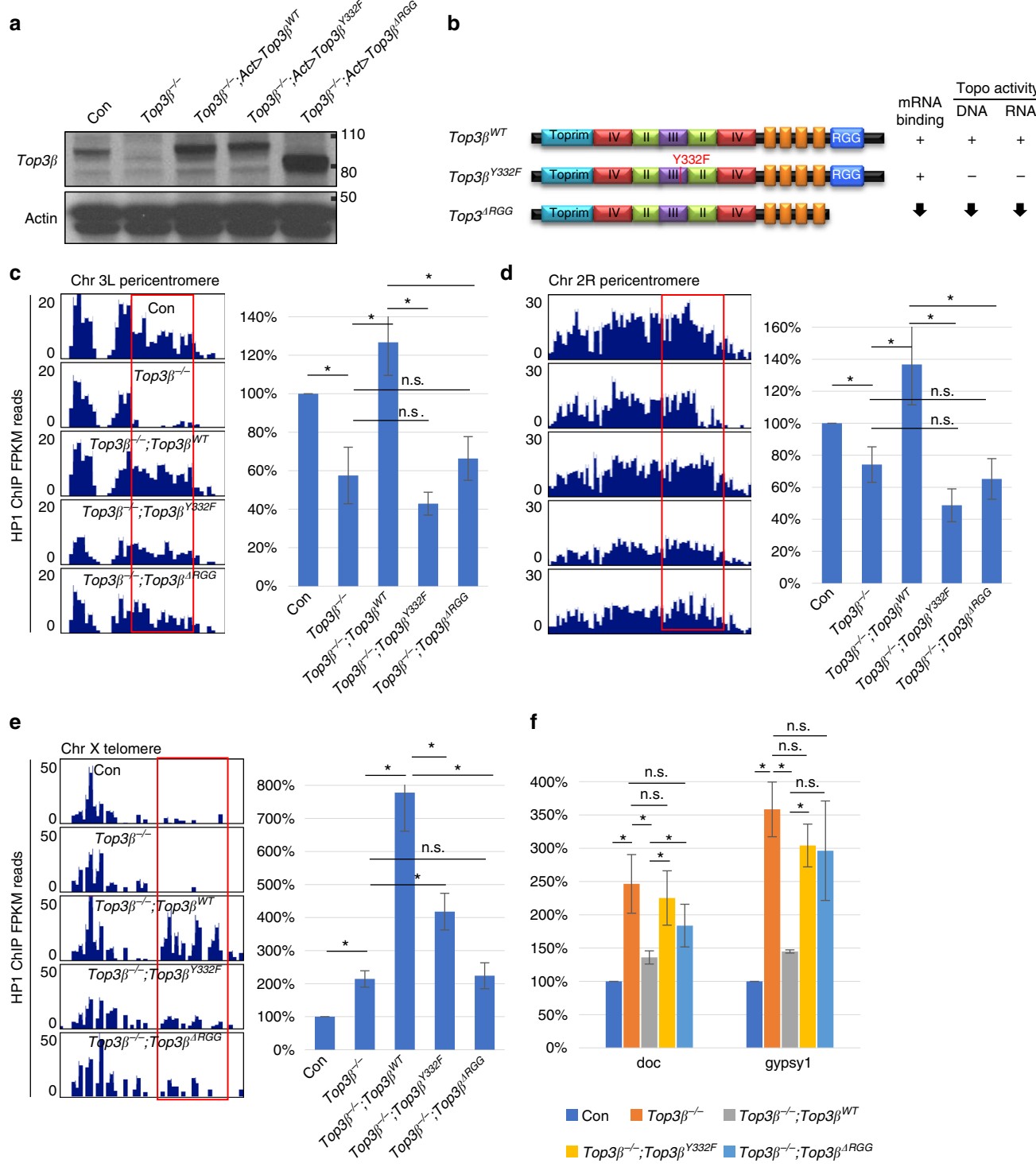

their targets by base-pairing interactions with nascent RNAs transcribed from the heterochromatin locus[37–39]. We predict that this process may produce RNA topological problems at some loci, which depend on Top3β to resolve. Possibly, the nascent RNA may generate complex structures by: base-pairing with itself, or other RNAs, or DNA. These structures may prevent the nascent RNAs from being base-paired by the siRNA-loaded RISC, so they need to be resolved by the topoisomerase activity of Top3β.

How may Top3β solve RNA topological problems during RISC-mediated heterochromatin formation? One RISC component that interacts with Top3β is the p68 RNA helicase, which

interacts with histone methyltransferase and is needed for heterochromatic gene silencing[19,40,41]. The biochemical and genetic interactions between a Type IA topoisomerase and helicase have been similarly observed for Top3α and BLM: their enzymatic activities are coupled to resolve complex DNA structures[42,43]. Based on this similarity, we hypothesize that Top3β and p68 may employ a mechanism mimicking their DNA counterpart—their enzymatic activities may be coupled to resolve complex RNA structures during heterochromatin formation (Fig. 9). p68 may unwind secondary structures in nascent RNA transcribed at heterochromatin. This unwinding may generate topological

**Fig. 8** Top3β depends on both RNA-binding and catalytic activity to recruit HP1 to specific loci in heterochromatin and silence TEs. **a** Western blotting of Top3β complementation in fly heads. Each transgene (listed in **b**) was expressed by actin-gal4 driver in $Top3\beta^{26}$ mutant background. **b** Schematic representation of the wildtype, the catalytic point mutant (Y332F), and the RGG-box deletion mutant (ΔRGG) of Top3β protein and their biochemical activities. The presence, absence, and reduction (arrow) of the activity are indicated. **c** Bedgraphs display HP1 ChIP-seq data at a representative locus in chromosome 3L pericentric region from the control flies ($w^{1118}$), Top3β mutant ($Top3\beta^{-/-}$), and the transgenic flies ($Top3\beta^{WT}$, $Top3\beta^{Y332F}$, and $Top3\beta^{\Delta RGG}$) that express wildtype and two mutants of $Top3\beta$, in the $Top3\beta$ mutant background. The red box indicates the region selected for ChIP-qPCR analyses of HP1. The right graph shows the validation of the ChIP-seq by ChIP-qPCR analysis. **d** Same as **c** except that the representative locus is from pericentric heterochromatin of chromosome 2R. Note that the reduction of HP1 ChIP signals are significantly decreased in $Top3\beta^{-/-}$ as compared to control, and also in mutant complementation as compared to $Top3\beta^{WT}$ complementation ($p < 0.05$). **e** Bedgraphs displaying FPKM of ChIP-seq data show that HP1 signals are increased in $Top3\beta^{WT}$ complemented $Top3\beta^{-/-}$ mutant in the telomere locus in chromosome X. The graphs on the right panel represent ChIP qRT-PCR of the respective locus. Note that wildtype $Top3\beta$ transgene ($Top3\beta^{WT}$) resulted in the increase of HP1 recruitment in $Top3\beta^{-/-}$ mutant in chromosome X telomere, whereas the catalytic mutant and RNA binding mutant transgenes induced lower levels of HP1 as compared to $Top3\beta^{WT}$. **f** A graph showing RT-qPCR of TE expression level in the control ($w^{1118}$), $Top3\beta^{-/-}$, $Top3\beta^{WT}$, $Top3\beta^{Y332F}$, and $Top3\beta^{\Delta RGG}$ complemented mutant. RT-qPCR was performed in quadruplicates and TE levels were measured by normalizing with rp49. Statistics in **c–f**: "n.s." indicates difference that is statistically not significant ($p > 0.05$); and asterisks indicate statistically significant difference with $p$-values less than 0.05 by Student $T$-test. The error bars represent standard errors

stress, which may be relieved by Top3β to enable p68 to continue the unwinding reaction, making nascent RNA accessible to binding by siRNA-loaded RISC, and enabling subsequent recruitment of histone methyltransferases and HP1. When either p68 or Top3β is mutated, this process is defective, leading to defective heterochromatin, and increased *white* gene expression in PEV assay. However, this hypothesis is difficult to explain why the double mutant between *Rm62* and *Top3β* exhibits suppression of the PEV phenotype observed in each single mutant. We speculate that when both p68 and Top3β are mutated, alternative pathways may be activated to form heterochromatin. One alternative pathway may be mediated by PIWI[34,44], which can interact with FMRP to promote heterochromatin silencing in somatic cells[45]. Future work is needed to show whether activities of Top3β and p68 are coupled, and to identify the alternative pathway(s).

Finally, our findings that Top3β depends on both RNA binding and catalytic activities for recruiting HP1 and silencing TEs are reminiscent of our previous data that Top3β requires the same two activities to promote synapse formation[8], suggesting that these Top3β-mediated processes may be connected. Indeed, misregulation of heterochromatin and TEs has been linked to abnormal brain development, schizophrenia, other neurological disorders[46], and aging[47]. Notably, individuals carrying *Top3β* mutations have increased risks of schizophrenia and other mental disorders[3,8]; and *Top3β* mutant animals exhibit abnormal neurodevelopment and shortened lifespan[4,48]. Thus, regulation of heterochromatin formation and transcription silencing could be one mechanism used by Top3β to prevent mental dysfunction and shortened lifespan.

## Methods

**Drosophila stocks and genetics.** Flies were cultured on corn syrup-soy recipe food from Archon Scientific at $25 \pm 1\ °C$ and $60 \pm 5\%$ humidity, under a 12 h/12 h light/dark cycle, except for the PEV assay, which was performed at 22 °C. $Top3\beta$ p-element remobilized mutants $Top3\beta^{26}$, $Top3\beta^{16}$, and $Top3\beta^{37}$ were a generous gift from T.S. Hsieh[49]. Su(var)3-9[906]/TM6, Dcr-2[R416X]/CyO, PEV line 39C-2, 39C-12, 118E-10, and 118E-28 were a generous gift from S. Elgin. $y^1 w/Dp(3;Y)BL2$, $P[HS\text{-}lacZ.scs]65E$, $w^{1118}$; $AGO2^{321}/TM3$, $Sb^1$, $T(2;3)Sb^V$, $In(3R)Mo$, $Sb^1$ $sr^1/TM3$, $Ser^1$, $cn^1$ $P[ry[+t7.2]]$ $=\{PZ\}AGO1^{04845}/CyO$; $ry^{506}$, $w^{1118}$;$P\{Flag.HA/AGO2\}2$ and $w^{1118}$; $Fmr1^{\Delta50M}/TM6B$, $Tb^1$ were obtained from the Bloomington Stock Center. $AGO2^{1\text{-}1\text{-}7}$ was generated by the standard CRISPR Cas9 targeting protocol[50]. The gRNAs are designed to target the promoter (GTGATGGGAGGCCTTAG) and exon 6 (GTCAGCCAC-CAGGCCATCC). Injection of the gRNA expressing plasmids was performed by Thebestgene injection service (https://www.thebestgene.com/). The screened and sequence verified $AGO2^{1\text{-}1\text{-}7}$ line contains a 630 bp deletion in the promoter region and a 7 bp-deletion in exon 6, creating a null-allele. $Top3\beta$ transgenic lines UAS-$Top3\beta^{WT}$, UAS-$Top3\beta^{Y332F}$, and UAS-$Top3\beta^{\Delta RGG}$ were described previously[8]. For $Top3\beta$ complementation, heads of $Top3\beta^{26}$; UAS-$Top3\beta$/Act-Gal4 were used in RT-qPCR and HP1 ChIP experiments.

**Cell culture, immunoprecipitation, and Western blot.** Schneider S2 cells (Invitrogen R69007, Fisher Scientific, Waltham, MA, USA) were cultured in Schneider's S2 media (Gibco, Thermo Fisher, Waltham, MA, USA) supplemented with 10% heat-inactivated fetal calf serum and 1% of penicillin and streptomycin (Invitrogen, Thermo Fisher, Waltham, MA, USA) at 25 °C. Tagged expression constructs (Flag full-length TDRD3, TDRD3-N250, N440, N500, N547, N720, ΔCTD, 251C, 441C, 501C, 548C, Δloop, ΔUBA, Δ241-440, Δ501-546, ΔTYD, E772K, Flag-Top3β, Flag-Top3β-ΔRGG, HA-p68, HA-VIG, HA-AGO2) were cloned in pMT/V5 vector according to standard cloning procedure. The genomic layouts of the constructs are pictured in Fig. 2b. Transfection of plasmids and siRNAs were carried out with the Calcium Phosphate Transfection kit (Invitrogen, Thermo Fisher, Waltham, MA, USA) following the manufacturer's protocol. IP and MS were performed using our published protocol[4].

The immunoprecipitated eluates were analyzed by standard Western blot analysis. Rabbit TDRD3 and Top3β antibodies (produced in-house)[5] were used in 1:1000 and 1:200 dilutions, respectively. α-FMRP antibody was obtained from abcam (ab10299) and used in a 1:1000 dilution. Guinea pig α-p68 antibody (1:3000) was a kind gift from A. Spradling[51]. Mouse α-AGO2 serum (1:1) was a kind gift from M. Siomi[52]. Uncropped images for Western blotting data are included in Supplemental Figure 9.

**ChIP-seq and ChIP RT-qPCR.** ChIP and ChIP-seq library generation was carried out by standard protocols[53,54]. Fly heads (100 µl) were isolated from fast freeze and sieving. The antibodies used are: HP1—Developmental Studies Hybridoma Bank (DSHB) C1A9; H3K9me2—Abcam ab1220; H3K9me3—Abcam ab8898; FLAG—Sigma-Aldrich F7425; Top3β—our own group[5]. The homogenized and cleared samples were sonicated to a length between 200 and 500 bp. The sheared chromatin was precleared with Protein A beads (GE Healthcare 17-1279-01, Waukesha, WI, USA), and then incubated with 5 µl of each of the above antibodies and 30 µl of Protein A beads at 4 °C for 3 h. After thorough washing, the samples were eluted by 10% SDS and Protease K at 65 °C for 8 h. The eluted chromatin was extracted by phenol–chloroform clean-up followed by ethanol precipitation. For RT-qPCR, 2 µl of 10× diluted chromatin was used for 20 µl reaction using ABI7800HT. Given the reaction was run in triplicate, the threshold values (Ct) were converted to fold change difference in standard delta Ct method. The primers used in the analysis are in Supplemental Table 1.

ChIP-seq libraries were sequenced by the Hi-Seq 2000 at 50 bases. Reads were mapped to the dm6 fly genome with Bowtie[55] allowing maximum of 2 mismatches. Non-unique reads with 2 or 3 hits in the genome were assigned weights of 1/2 or 1/3, respectively. Reads with >3 hits in the genome were not counted. The frequency of reads was calculated in 100-bp intervals across the genome in FPKM units. As a control, we used total DNA input samples for the same line of flies ($w^{1118}$ or mutants), that were processed in the same way as the ChIP-seq samples. Input-subtracted positive ChIP-seq frequencies were visualized using the UCSC Genome Browser[56]. For Flag-AGO2 ChIP analysis, the background noises were removed by subtracting Flag ChIP-seq signals from $w^{1118}$ control flies which do not express any Flag-tagged proteins. This subtraction is necessary because Flag ChIP-seq produces high background signals, which interfere with the real Flag-AGO2 ChIP signals. It is known that highly expressed loci are non-specific "hotspots" for ChIP-seq that may lead to misleading localization of proteins on chromatin by this technique[57]. A large amount of Flag-ChIP signals from $w^{1118}$ control line are from transcription start sites of active genes in euchromatin, so that they are likely derived from the non-specific "hotspots".

For SICER analysis, the ChIP-seq results are quantitated by SICER 1.1 against *Drosophila* genome version dm6, with windows of 100 bp-width and 600 bp-gap. The regions with significant SICER scores obtained were referred to as "islands". The scores for each mutant and control ($w^{1118}$) were subtracted of their respective input signals. False discovery setting is <5%, and islands with scores less than 200

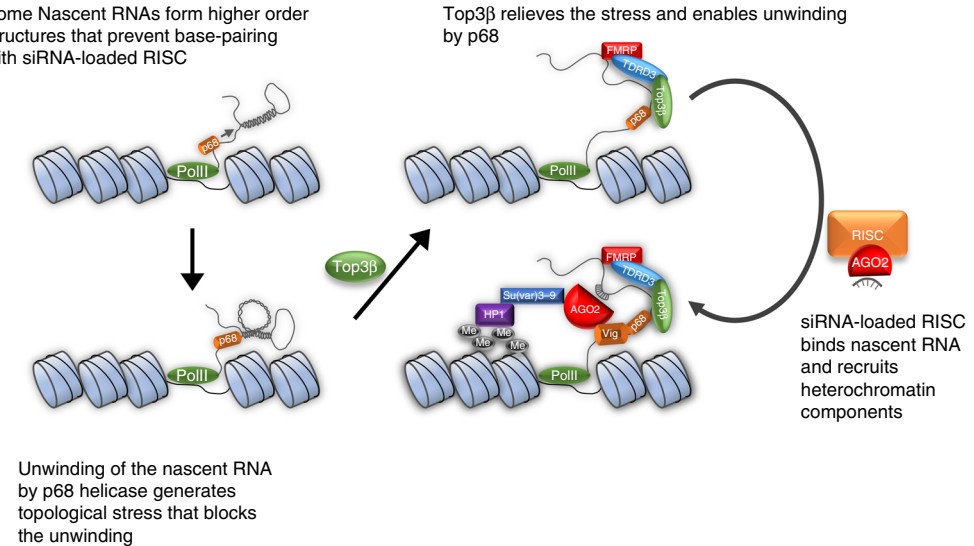

**Fig. 9** Top3β promotes heterochromatin formation by interacting with RNAi machinery. A model illustrating how Top3β works with RISC in heterochromatin formation and transcriptional silencing. The nascent RNA transcribed from heterochromatin may form secondary and other structures by base-pairing with itself, which prevents access by siRNA-loaded RISC. These RNA structures are unwound by p68 helicase of RISC, and this process may produce torsional stress. Top3β may relieve this stress by facilitating the unwinding process and makes nascent RNA accessible to base pairing by siRNA-loaded RISC. Binding of RISC to the nascent RNA may recruit HP1 and H3K9 methyltransferases to assemble heterochromatin. The coordinated actions of p68-Top3β in resolving complex RNA structures may resemble those of their homologs, BLM-Top3α, in resolving complex DNA structures

are excluded from further analyses. The input-subtracted scores for each mutant were compared with those of the control ($w^{1118}$) by using the bedtools suite *intersect*, with settings of 50% cross and 200 cut-off. Three categories of ChIP islands were obtained: the mutant-loss islands, mutant-obtained islands, and the mutant/wildtype-shared islands; in which the ChIP signals are decreased, increased, and of no change, respectively. These islands were combined to produce the bedgraph format files, which were loaded onto genome browser for manual analysis of ChIP signals for their chromosome location, base coverage area, relative levels, and distributions. Comparison of data of 3 samples or more used *bedtools multiIntersectBed* with the same cross criteria and cutoff score.

For quantification, SICER signals from pericentric heterochromatin and euchromatin of chromosome 2 and 3 were separately calculated for the percentage of islands that are decreased, increased, or of no change in each mutant comparing to the control. The total length covered by each type of islands was also calculated for pericentric heterochromatin. The pericentric heterochromatin regions are distinguished from euchromatin based on chromosome band, which is largely consistent with our ChIP-seq data of HP1. They were manually set as: 2L: 22,090,000–23,513,711; 2R: 1–4,505,672; 3L: 23,247,000–28,110,226; 3R: 1–4,165,641. The remaining regions on the two chromosomes were treated as euchromatin in calculations. The quantification of HP1 data is based on average of SICER scores from 3 independent ChIP experiments for *Top3β*, 2 for *AGO2*, and 2 for *Top3β/AGO2* mutants. The quantification of H3K9me2 and H3K9me3 data is based on SICER analysis from 3 independent ChIP experiments.

The overlapping of ChIP islands between HP1, Top3β, and AGO2 was calculated based on the width of ChIP SICER peaks. To summarize, both heterochromatin and euchromatin SICER-predicted islands were selected depending on the subject. The overlaps between two ChIP SICER scores were obtained by crossing the two ChIP SICER islands using *bedtools intersect* with at least 50% overlap of the shortest island. The uncrossed portions were obtained by subtracting the crossed SICER island's width from the total islands of either heterochromatin or euchromatin. The percentage of the SICER islands of a special protein in total heterochromatin was calculated by using the width of SICER islands divided by the length of total heterochromatin of chromosome 2 and 3 (14,956,269 bases).

**RNA-sequencing and microarray analysis**. A total of 200 µl *Drosophila* heads/per sample were collected by fast freezing and sieving. Total RNA was extracted using TRIzol reagent according to the manufacturer's protocol (Thermo Fisher, Waltham, MA, USA). Three sets of independently prepared RNA samples were applied for RNA sequencing and microarray assay. For the microarray, RNA concentration and quality were measured by Nanodrop (Thermo Fisher, Waltham, MA, USA) and the Agilent Bioanalyzer RNA 6000 Chip (Agilent, Santa Clara, CA). Two hundred nanograms of total RNA was labeled using the Agilent Low-Input QuickAmp Labeling Kit, and was purified and quantified according to the manufacturer's recommendations. A total of 600 ng Cy3-labeled cRNA was hybridized for 17 h to Agilent Unrestricted AMADID Release GE 4x44K 60mer *Drosophila* V2 oligo microarrays (G2519F). Following post-hybridization rinses, arrays were

scanned using an Agilent SureScan microarray scanner at 3 micron resolution, and hybridization intensity data was extracted from the scanned images using Agilent's Feature Extraction Software.

Microarray data was analyzed using parametric statistical analysis approach and the JMP11 platform-based DIAINE 6.0 analysis suit. After normalization of the gene expression intensities by $z$-transformation and generation of $z$-scores, samples were first examined by sample scatter plots, sample hierarchical clusters, and principal component analysis. Possible outlier samples were excluded using pairwise analysis. Then, pairwise analysis between different gene groups were performed by pairwise $z$-test with multi-comparison correction to generate $z$-test $p$-value, $z$-ratio, fold-change, false discovery rate (FDR), and average gene expression levels for each comparison group. All microarray results were filtered globally by selecting probes with probe detection $p$-values ≤ 0.02, and sample group one-way independent ANOVA $p$-values < 0.05. The results from probes not meeting the above criteria were excluded. The filtered probes were then further selected to create a list of significant probes that must pass all four criteria for each comparison: (a) $z$-test $p$-values < 0.05, (b) FDR ≤ 0.30, (c) $z$-ratio or fold change absolute values >=1.5, (d) average signal intensity for each comparison group ≥0. Only results from the list of significant probes were considered significant and used to produce the significant gene lists. When a gene has multiple significant probes, the maxium of the average signals of all probes was used as the levels of gene expression in subsequent calculations. The difference between levels of gene expression was then calculated, and the data obtained were utilized for gene set enrichment analysis with the *Drosophila* gene oncology database to find the significant functional/pathway level changes and gene–gene interactions.

RNA-seq analysis was proceeded by genome mapping on dm6 version *Drosophila* standard genome by Tophat 2.0.9 with gene coordinators as template and standard parameters. The mapping result bam format files were aligned using Tuxedo tool suite and generated common alignment transcript files and each sample's FPKM data for further study. The change of gene expression was calculated by Cuffdiff from Tuxedo tool suite and then summarized by mapping gene position. Largest transcript changes for each gene was selected with a $p$-value and $q$-value. The gene list was first filtered by ANOVA test on $z$-scores with $p$-value ≤ 0.05. Then the significant genes were selected on the filtered gene by (a) $q$-value ≤ 0.05, (b) fold change value ≥ 1.5. The sample correlation studies were performed by using FPKM of each gene as the raw gene signal and performed the log $z$-transform to obtain gene expression $z$-score as the normalized gene expression level for each gene on each sample. The Pearson correlation coefficients were analyzed and combined with the agglomerative hierarchical clustering approach with average distance method which was used to find possible sample group outliers. This group aggregation effect was further examined by principal component analysis to unveil the group separation patterns. The gene expression data were further used as the input to calculate gene set enrichment analysis by PAGE algorithm (https://bmcbioinformatics. biomedcentral.com/articles/10.1186/1471-2105-6-144) to monitor relative functional change with various comparison pairs.

**Immunostaining.** The wandering third instar larvae were washed in 1× PBS and dissected for the salivary glands. The glands were fixed in 4% paraformaldehyde (pH 7.6) for 20 min at room temperature. After washing four times, the samples were incubated in 0.1% Triton X100 and 1% normal goat serum in 1× PBS for 30 min at room temperature for permeabilization. The primary antibodies (α-mouse HP1—DSHB C1A9 1:250; α-mouse H3K9me2—abcam ab1220 1:200; α-rabbit H3 —abcam ab1791 1:200) and secondary antibodies (1:400) were incubated in 0.1% Triton X100 and 1% NGS overnight at 4 °C. Washing was performed three times in 10 min intervals after each incubation with 0.1% Triton X100 and 1% NGS. Preparations were examined with Zeiss LSM-710 confocal laser scanning microscopy. For quantification of HP1 and H3K9me2, respective channels were separated and the intensity per area was measured using ImageJ.

**PEV, eye pigment, LacZ, and bristle quantification.** PEV assays using different *white* reporters have been described[18,58,59]. All crosses were performed at 25 °C except for $w^{m4h}$ at 22 °C. The phenotypes were examined 48 h after eclosion. For eye color quantification, 10 fly heads were homogenized in 0.1% HCl in methanol. After incubating overnight at 4 °C, the absorbance of the cleared supernatant was measured at 480 nm. The data were obtained from at least three independent experiments.

The PEV assay with a LacZ reporter was performed using a published protocol[60]. To summarize, crawling 3rd instar larvae were heat-shocked at 37 °C water bath for 30 min followed by recovering at room temperature for 30 min. The larvae were dissected in PBS and salivary glands were fixed at 0.2% glutaraldehyde for 20 min at room temperature. After washing three times with PBS, the samples were incubated with 1 mg/ml X-gal for 30–60 min. Stained samples were mounted and imaged with Leica M165FC microscope. The assay was repeated three times and at least 10 larvae per sample were analyzed each time.

The PEV assay based on the *Sb* reporter was performed using a published protocol[61]. $w^{1118}$ and $Top3\beta^{26}$ females were crossed to male $T(2;3)Sb^V$, $In(3R)Mo$, $Sb^1$ $sr^1/TM3$, $Ser^1$, $cn^1$ to assay $Sb^V$ variegation modification. The male progenies were analyzed by counting 7 pairs of major dorsal macrochaetes. The assay was done three times counting at least 20 flies for each trial.

The RNAi-mediated post-transcriptional silencing assay was performed using a published protocol[25]. The reporter line (GMR-wIR), which expresses an inverted repeat targeting *white*, was crossed with $Top3\beta^{26}$ or *Dcr-2* mutant, and the eye phenotype was examined for more than 20 flies in each genotype.

## Data availability

All relevant data supporting the key findings of this study are available within the article and its Supplementary Information files or from the corresponding author upon reasonable request. The next-generation sequencing and microarray data have been deposited at GEO database (GSE119736 and GSE119185).

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

## Acknowledgements

We thank Drs. S. Elgin for PEV lines; T.S. Hsieh for *Drosophila Top3β* flies and antibody; A. Spradling and M. Buszczak for p68 antibody and fly lines; S. Artavanis-Tsakonas and R. Obar for p68 and Vig plasmids; H. Siomi for reagents of AGO2, FMRP, and p68; E. Izaurralde for AGO2 plasmid; S. Hammond and K. Ivanov for Vig reagents; Y. Lee and R. Carthew for GMR-white IR fly lines. This work is supported in part by the Intramural Research Program of the National Institute on Aging (Z01 AG000657-08) and the Intramural Research Program of the National Institute of Diabetes and Digestive and Kidney Disease (ZO1 DK015602-09), National Institutes of Health.

## Author contributions

S.K.L., W.S., S.Z., and W.W. designed experiments. S.K.L., Y.X., W.S., Y.J., M.A., M.C., Y. D., and E.L. performed experiments. S.K.L., Y.Z., W.L.K., S.D., A.S., and W.W. analyzed the data. Y.D., W.L.K., K.G.B., E.P.L., K.Z., S.Z., A.S., and W.W. supervised the project. S. K.L. and W.W. wrote the manuscript with the input from all authors.

## Additional information

**Competing interests:** The authors declare no competing interests.

