## [Peer Review File · Nature Communications]

Reviewer #1 (Remarks to the Author):

Topoisomerases are essential enzymes involved in regulation of the topological states of double-stranded nucleic acids, DNA in most cases. Recent studies have identified a unique class of dual-activity DNA/RNA topoisomerase Top3 β , but its biological functions remain unclear. In this paper, authors present extensive biochemical and genetic data that collectively demonstrate that Top3 β interacts with RNAi-induced silencing complex (RISC) components and thereby contributes to the establishment of heterochromatin in *Drosophila*. The work identifies important insights into the novel role for an RNA topoisomerase in gene regulation. I support publication of this paper, although I have a couple of reservations as listed below.

- It is unclear whether the observed requirement for Top3 β in heterochromatin formation is actually dependent on the topoisomerase activity of Top3 β and, if it is, whether it's the DNA or RNA topoisomerase activity that's important. Several mutant strains are used in the genetic studies but they are with deletion of the top3 β gene. Examining the effect of mutations of the topoisomerase catalytic residues (e.g. phenylalanine substitution for the catalytic tyrosine) and/or RNA binding RGG domain would address this critical point.

- The data showing physical interaction between the TDRD3-Top3 complex and RISC are weak, with bands for the RISC components not detectable in the silver-stained gel of TDRD3 immunoprecipitates in Fig. 1. Authors use the number of peptides identified in MS analysis as the evidence of complex formation, but it's likely that peptides from many other unrelated proteins were also detected, and they should provide information as to how selective the reported interactions were. In addition, the authors also use the number of peptides as the basis for comparing the strengths of interactions in several contexts (e.g. in the presence vs. absence of RNase). Although the number of recovered peptides probably reflects abundance, I don't think this is a reliable way for quantitative comparison. Comparing band intensities in western blots, for instance, would seem more appropriate.

Reviewer #2 (Remarks to the Author):

In this manuscript Lee and co-workers present very interesting findings reporting the role of Topoisomerase 3 β in heterochromatin formation and transcriptional silencing. The role of Top3 β

has only recently started to be elucidated and this protein is currently receiving significant attention given its link to autism and mental retardation. Dissecting the molecular pathways in which Top3 β is involved is thus of prime and wide interest.

So far the data highlights that Top3 β and its interaction with FMRP are mostly related to the regulation of mRNAs translation. In this manuscript, a novel mechanism is proposed, namely the direct modulation of heterochromatin formation and transcriptional silencing. The findings are thus novel, important and with potential great impact.

However, the manuscript falls short in explaining how exactly Top3 β does its job. Considering the “dual activity” of this enzyme, this manuscript would greatly benefit from a deeper analysis on which activity is involved. Moreover, there seems to be a disconnection from the biochemical data and the functional one. The manuscript provides beautiful biochemical characterisation of the interactions via TDRD3 but it was never tested how such mutant/truncated forms perform with regard to PEV (or whether TDRD3 mutants also modulate PEV). Top3 β mutant for the RGG box equally bind RISC complex. Do these mutants also alter PEV? Further dissection on the actual role of Top3 β may shed light on the antagonistic effect of Top3 β and RISC that the authors propose.

Indeed, the antagonistic effect between Top3 β and RICS complex reported is well supported by the data but the complex epistasis observed is not fully exploited. It remains unclear, even with the speculative model presented in figure 7, how such antagonistic effect could be explained. The authors state this genetic interaction suggests “Top3 β works together with siRNA-loaded RISC” but more evidence would be needed to support this hypothesis.

In summary, I found the manuscript very interesting, with a compelling amount of data, the experiments were properly conducted and the presentation is logical and clear. But it still presents some gaps to fully understand the role of Top3 β in this novel function.

Other specific comments:

- 1) The title is somehow misleading as it places emphasis on the “dual activity” of Top3 β . In fact, throughout the manuscript, it remains unclear which activity is involved in the reported findings. Maybe “Topoisomerase 3 β interacts with RNAi machinery...” would work best.
- 2) The finding that the interactions are mostly kept with Top3 β RGG-box mutant raises some doubts as to which activity can be used here? Did the authors ever test the effect of RGG mutations in PEV?

3) The antagonistic role in PEV for the reported interaction is the less clear part of the entire manuscript. The evidence that Top3 β mutations on their own lead to a reduction in silencing is quite clear and well documented. But the findings arising from the double mutants (together with RISC components) are less clear to understand. How exactly can Top3 β mutations bypass the need for RISC complex in chromatin silencing? Extensive and compelling evidence supports such antagonistic role throughout the manuscript (e.g. genetic evidence, ChIP for heterochromatin markers, HP1 localization etc.) so it is certainly occurring. But the epistasis of this interaction is rather complex and not very clear to understand at the mechanistic level. The discussion of putative models (as presented in figure 7) is also lacking an explanation for the interactions observed.

4) The finding that Ago binding to Top3 β , unlike other RISC components, is not RNase sensitive (Sup) is quite intriguing. But the authors do not even mention this in the results description. Can the authors comment on that?

5) The conclusion that "RISC plays more important roles than Top3 β in HP1 recruitment" (page 11) needs additional data to support the claim. Comparative analysis of quantitative value arising from different are rather tricky to evaluate and the authors may wish to consider keeping these comparative approaches as a description note, rather than a stand alone conclusion. (on a side note, the excessive use of latter/former in the first paragraph makes it very difficult to read...)

6) The data for gene expression at the sub-telomeric regions is very interesting (figure 7). Although the HP1 binding suggest that these regions are not subjected to the same antagonistic role between Top3 β & RISC (fig.5), it would be interesting to test at the transcript level is any change is detected in the double mutants.

7) The authors should state the type of alleles used (null, truncations, point mutations, etc) in the M&M rather than simply referring to the original source.

Reviewer #3 (Remarks to the Author):

In the manuscript "A Dual-Activity Topoisomerase Interacts with RNAi Machinery to Promote Heterochromatin Formation and Transcriptional Silencing" by Lee et al., the authors studied the Topoisomerase 3 β role in its interaction with RNA and the RISC complex as well as its relevance in transcriptional silencing maintenance.

This work provides important novelties in the field. Beside to confirm in *Drosophila* the Topoisomerase 3 β interaction with TDRD3 already observed in human [Xu et al., *Nat Neurosci.* 2013; 16(9):1238-47], it shows the interaction with the RISC complex on biochemical basis. In addition, the

biochemical and genetic data concerning the transcriptional silencing maintenance involving TDRD3, are important and innovative.

The involvement of Topoisomerase 3 β in the epigenetic control of transcription may overcome the specific interest of the field.

The data presented are based on well performed experiments and most of them sound with the conclusions of the manuscript.

However, in order to satisfy the standard quality of Nature Communications, I would suggest some specific modifications and additional experiments:

#1. This work underlines the Topoisomerase 3 β role in transcriptional silencing control. From the data presented, it is not evident whether this control derives from the Topoisomerase 3 β catalytic activity or from its sole presence (scaffold activity). In order to clarify this point, authors should provide experiments, analogous to those presented in Fig. 3 and 4, studying a Topoisomerase 3 β mutant in which the catalytic tyrosine (Ahmad et al., *Nucleic Acids Res.* 2017 17;45:2704-2713) is replaced with phenylalanine at position 322 (Y322F) to show how Topoisomerase 3 β behaves in these controls.

#2. Authors claim that Topoisomerase 3 β , together with RISC, promotes HP1 recruitment to pericentric heterochromatin. This is mostly derived from ChIP seq results obtained with antibodies against HP1 in Topo3 β ^{-/-} or Ago2^{-/-} mutants. A ChIP-seq using antibodies against Topoisomerase 3 β and AGO2 should be performed in order to show a Topoisomerase 3 β and RISC presence overlapping to that of HP1. This would more directly show the point.

Minor modifications required.

Top3 β and RISC seem to contribute to the recruitment of HP1 to pericentric and telomeric heterochromatin. However their interaction is antagonistic only for pericentric but not for telomeric heterochromatin: the title of this latter paragraph (pag 12, line 308) should be corrected to emphasize this conclusion.

Figure S4 shows distribution of H3K9 methylation and HP1 presence in WT and Topo3 β ^{-/-} mutant cells. The reduction of the H3K9me2 and HP1 signals in Topo3 β ^{-/-} mutant should be presented as merged images in order to confirm the effect of Topo3 β ^{-/-} mutation.

There is a quite extended series of data concerning HII sites observed in euchromatin regions. This seems to be a minor effect of the processes described. Actually, authors never mention it in the discussion. To move these data to supplemental informations would be more appropriate.

Response to reviewers

Reviewers' comments:

Reviewer #1 (Remarks to the Author):

Topoisomerases are essential enzymes involved in regulation of the topological states of double-stranded nucleic acids, DNA in most cases. Recent studies have identified a unique class of dual-activity DNA/RNA topoisomerase Top3 β , but its biological functions remain unclear. In this paper, authors present extensive biochemical and genetic data that collectively demonstrate that Top3 β interacts with RNAi-induced silencing complex (RISC) components and thereby contributes to the establishment of heterochromatin in drosophila. The work identifies important insights into the novel role for an RNA topoisomerase in gene regulation. I support publication of this paper, although I have a couple of reservations as listed below.

- It is unclear whether the observed requirement for Top3 β in heterochromatin formation is actually dependent on the topoisomerase activity of Top3 β and, if it is, whether it's the DNA or RNA topoisomerase activity that's important. Several mutant strains are used in the genetic studies but they are with deletion of the top3 β gene. Examining the effect of mutations of the topoisomerase catalytic residues (e.g. phenylalanine substitution for the catalytic tyrosine) and/or RNA binding RGG domain would address this critical point.

Our Response: We thank reviewer 1 for his/her support in publishing our manuscript.

Reviewer 1 and the other two reviewers have each suggested that we examine whether a point mutation of the Topoisomerase catalytic residue (Y332F) and deletion of the RNA binding RGG domain (Δ RGG), can disrupt the activity of Top3 β in promoting heterochromatin formation. In the revised manuscript, we added several new figures to show that both mutations disrupt the ability of Top3 β in promoting HP1 recruitment to specific heterochromatin loci, and to silence specific transposable elements. These new experiments are based on the transgenic fly lines previously constructed by our group to show that both Y332F and Δ RGG mutations disrupt the ability of Top3 β to promote synapse formation at the neuromuscular junctions (Ahmad et al., NAR 2017). These lines express either the Top3 β -wildtype, Y332F, or the Δ RGG mutant in the Top3 β -null background (new Figure 8). The new data from analyses of these lines are summarized below:

1. Our HP1 ChIP assays show that the two previously identified HP1-reduced loci in pericentric heterochromatin of chromosomes 2R and 3L are rescued by transgenic expression of the wildtype Top3 β protein, but not its two mutants (Fig. 8C, D).
2. Our ChIP assays also showed that transgenic expression of wildtype Top3 β protein can induce an increase of HP1 to a level higher than that of the control line (w^{1118}) at a locus in telomeric regions of chromosome X, whereas its two mutant proteins are deficient in this assay (Fig. 8E)
3. We have used RNA-seq to analyze two transposable elements (TEs) located in pericentric heterochromatin, and have identified two transposable elements (TEs) that are consistently de-silenced in all Top3 β mutant fly lines (Fig. 7D, E). Interestingly, transgenic expression of wildtype type Top3 β repressed the levels of these TEs, whereas expression of the two mutants are deficient in repression (Fig. 8F).

Together, our new data suggest that Top3 β depends on its RNA binding and catalytic activity to promote heterochromatin formation at specific loci and to silence specific TEs. Interestingly, these data parallel our previous findings that Top3 β depends on the same two activities to promote neurodevelopment (Ahmad et al., NAR 2018), implying that misregulated heterochromatin and TE silencing may contribute to the abnormal neurodevelopment in Top3 β mutant. Indeed, misregulation of heterochromatin and mobile elements have been previously connected to abnormal brain development, schizophrenia and other neurological disorders (Erwin et al., Nat. Rev. Neuro, 2014). Because Top3 β mutation has been linked to schizophrenia, our findings suggest that one potential pathological mechanism of Top3 β in mental retardation is disrupting heterochromatin and transposon silencing. We thank all reviewers for proposing this experiment, and have added these discussions to the text.

- The data showing physical interaction between the TDRD3-Top3 complex and RISC are weak, with bands for the RISC components not detectable in the silver-stained gel of TDRD3 immunoprecipitates in Fig. 1. Authors use the number of peptides identified in MS analysis as the evidence of complex formation, but it's likely that peptides from many other unrelated proteins were also detected, and they should provide information as to how selective the reported interactions were. In addition, the authors also use the number of peptides as the basis for comparing the strengths of interactions in several contexts (e.g. in the presence vs. absence of RNase). Although the number of recovered peptides probably reflects abundance, I don't think this is a reliable way for quantitative comparison. Comparing band intensities in western blots, for instance, would seem more appropriate.

Our Response: We agree with the reviewer that we were only able to detect FMRP, but not other RISC components in the silvered-gel of TDRD3 immunoprecipitate. However, we think that this does not necessarily mean that their physical interaction is weak. From our past experience, the likely interpretation is that only a minor fraction of TDRD3-Top3 β complex associates with RISC, whereas the majority of TDRD3-Top3 β does not. One can actually enrich this minor fraction of the complex by fractionation prior to immunoprecipitation (IP), to detect these minor components (see examples in Meetei et al., MCB 2003; Nature Genetics, 2003). However, we did not perform this experiment for the current paper because we feel that the evidence for Top3 β -TDRD3-RISC complex is already very strong, as summarized below.

First, we believe that the best evidence for complex formation is reciprocal IP-MS or IP-Western. The rationale is that when two proteins are present in the same complex, the antibodies against each protein should co-immunoprecipitate both proteins. We prefer MS over Western, because MS is unbiased, and can detect all proteins in the IP; whereas Western depends on the specificity of the antibody, and can often produce false-positive results because of the antibody cross-reactivity. In our paper, we have provided reciprocal IP-MS data for several RISC components, including FMRP and two p68 variants. Moreover, we cited the published AGO2 and Vig IP-MS data. All these reciprocal IP-MS data contain peptides from Top3 β and TDRD3 (Fig. 1A). Second, we have provided reciprocal IP-Western data (Fig. 1B and 1C), which are consistent with IP-MS data. Third, we have provided IP-MS data using Flag-tagged Top3 β and TDRD3 (Figure S1A and S1B), which also show the association between RISC and Top3 β -TDRD3. Notably, our negative control for this experiment— Mock IP from S2 cells that do not

express Flag-tagged Top3 β or TDRD3—failed to detect any RISC components. The results suggest that the association between Top3 β -TDRD3 and RISC must be specific (otherwise, RISC should be present in the Mock IP). Fourth, we have performed IP-western using various deletion mutants of Flag-TDRD3, which showed domain-specific association between TDRD3 and different RISC components (Fig. 2, S2). In our opinion, these data provide strong evidence for the physical association between Top3 β -TDRD3 and RISC.

To address the reviewer's concerns, we have modified our text to indicate that RISC components are largely undetectable by silver-staining analysis in TDRD3 immunoprecipitates, which suggest that only a minor fraction of Top3 β -TDRD3 associates with RISC. We added the statement indicating that the reported association between RISC and Top3 β -TDRD3 is specific, as none of the RISC proteins was isolated by Mock IP using the Flag antibody from S2 cells lacking Flag-Top3 β or Flag-TDRD3.

Regarding interpretation of MS data, our experience with MS is consistent with the reviewer's comment that the number of peptides recovered from MS often reflect the protein abundance, but they are not reliable for quantitative comparisons. We have added this precautionary note in our revised manuscript. We also emphasized in the text that we have performed IP-Western to verify the IP-MS data.

Reviewer #2 (Remarks to the Author):

In this manuscript Lee and co-workers present very interesting findings reporting the role of Topoisomerase 3 β in heterochromatin formation and transcriptional silencing. The role of Top3 β has only recently started to be elucidated and this protein is currently receiving significant attention given its link to autism and mental retardation. Dissecting the molecular pathways in which Top3 β is involved is thus of prime and wide interest.

So far the data highlights that Top3 β and its interaction with FMRP are mostly related to the regulation of mRNAs translation. In this manuscript, a novel mechanism is proposed, namely the direct modulation of heterochromatin formation and transcriptional silencing. The findings are thus novel, important and with potential great impact.

However, the manuscript falls short in explaining how exactly Top3 β does its job. Considering the “dual activity” of this enzyme, this manuscript would greatly benefit from a deeper analysis on which activity is involved. Moreover, there seems to be a disconnection from the biochemical data and the functional one. The manuscript provides beautiful biochemical characterisation of the interactions via TDRD3 but it was never tested how such mutant/truncated forms perform with regard to PEV (or whether TDRD3 mutants also modulate PEV). Top3 β mutant for the RGG box equally bind RISC complex. Do these mutants also alter PEV? Further dissection on the actual role of Top3 β may shed light on the antagonistic effect of Top3 β and RISC that the authors propose.

Our Response: We thank reviewer 2 for his/her positive comments on our manuscript.

As requested by reviewer 2 and the other reviewers, we have added a more detailed analysis to address the question which activity of Top3 β is required for heterochromatin formation, using transgenic flies expressing wildtype and mutant versions of Top3 β (Y332F and Δ RGG) for HP1 CHIP-seq and transposon de-silencing analyses. The new data showed that Top3 β depends on both its RNA-binding and

topoisomerase activities in promoting HP1 to specific loci in pericentric heterochromatin and silencing of specific transposons (See Response to reviewer 1).

We thank reviewer 2 for positive comments on our characterization of biochemical interactions between different domains of TDRD3 and RISC. We included these data in this paper to provide more biochemical evidence to convince readers (including Reviewer1) that *Drosophila* Top3 β -TDRD3 interacts with not only with FMRP, but also RISC. The data also serve the purpose to reveal similarity and difference between human and *Drosophila* Top3 β -TDRD3 complex regarding how they interact with FMRP and other factors. We completely agree with Reviewer 2 that it will be important to characterize the functional relevance of different domains of TDRD3. In fact, we have successfully made a CRISPR-KO of *Tdrd3* mutant, and found that it can modify PEV (data not shown). However, this study has been time-consuming, and requires generation of multiple new knockout and/or transgenic lines. The current manuscript focuses on Top3 β and already has 9 figures, and is also over the limit in terms of the word count. We plan to publish an independent paper in the future that specifically focuses on functional characterization of TDRD3.

Regarding PEV assay for *Top3 β* Δ RRGG mutant: this experiment requires generation of new fly strains. Because we have already generated transgenic lines of different *Top3 β* mutants, and used them to show that Top3 β depends on its RNA binding and catalytic activities to promote synapse formation (Ahmad et al., NAR 2017), we chose to analyze these lines by HP1 ChIP and transposon assays. One advantage of this approach is that the new data can be linked with our previous findings from the same flies, as they show that Top3 β depends on the same two activities for HP1 recruitment to specific heterochromatin loci and silencing of specific transposons (see our response to Reviewer 1).

Indeed, the antagonistic effect between Top3 β and RICS complex reported is well supported by the data but the complex epistasis observed is not fully exploited. It remains unclear, even with the speculative model presented in figure 7, how such antagonistic effect could be explained. The authors state this genetic interaction suggests “Top3 β works together with siRNA-loaded RISC” but more evidence would be needed to support this hypothesis.

Our Response: We agree with the reviewer that it is not easy to explain the antagonistic genetic interaction between *Top3 β* and RISC, which was observed in our experiment. To fully understand how they work together, one will need to do biochemical experiments *in vitro*. However, such experiments will require design and creation of novel RNA substrates suitable to reveal the coordinated actions of the p68 helicase, topoisomerase, and siRNA-loaded AGO2. We believe that this experiment should be done as an independent study in the future. As an example, a genetic study showed that Top3 α antagonistically interacts with the sgs1 helicase (BLM homolog in yeast) in 1994 (Gangloff S. et al., MCB 2004). But the mechanism of how Top3 α works with BLM was elucidated 9 years later in an elegant biochemical study using a double-Holliday junction DNA substrate (Wu and Hickson, Nature 2003). A similar effort will be needed to elucidate how Top3 β works with RISC.

In summary, I found the manuscript very interesting, with a compelling amount of data, the experiments were properly conducted and the presentation is logical and clear. But it still presents some gaps to fully

understand the role of Top3 β in this novel function.

Our response: We thank the reviewer for his/her positive comments, and agree that there are some gaps that need to be filled in. Our study is the first one that reveals a novel role of Top3 β in heterochromatin formation in conjunction with RISC. The revised manuscript also shows that Top3 β depends on both its RNA-binding and catalytic activities to promote heterochromatin formation and silencing of transposons. The data provide one potential mechanism on how Top3 β may contribute to the defective neurodevelopment, mental dysfunction, and shortened life-span, observed in patients and animal models with *Top3 β* mutants. The detailed mechanism on how Top3 β works in this process can be addressed by future studies using biochemistry and other approaches.

Other specific comments:

1) The title is somehow misleading as it places emphasis on the “dual activity” of Top3 β . In fact, throughout the manuscript, it remains unclear which activity is involved in the reported findings. Maybe “Topoisomerase 3 β interacts with RNAi machinery...” would work best.

Our Response: We have revised the title as suggested.

2) *The finding that the interactions are mostly kept with Top3 β RGG-box mutant raises some doubts as to which activity can be used here? Did the authors ever test the effect of RGG mutations in PEV?*

Our Response: As mentioned in our response to the first comment by Reviewer 1 and 2, we have used transgenic rescue experiments to show that *Top3 β* mutant deleted of its RGG-box is deficient in promoting HP1 recruitment to specific loci in pericentric heterochromatin and in silencing of several transposons. We did not perform the PEV assay, because this requires generation of new mutant lines. One advantage of using the transgenic lines is that we can compare our new data with the old ones (Ahmad et al., NAR 2017) to link defective heterochromatin formation and transposon silencing with neurodevelopment.

3) *The antagonistic role in PEV for the reported interaction is the less clear part of the entire manuscript. The evidence that Top3 β mutations on their own lead to a reduction in silencing is quite clear and well documented. But the findings arising from the double mutants (together with RISC components) are less clear to understand. How exactly can Top3 β mutations bypass the need for RISC complex in chromatin silencing? Extensive and compelling evidence supports such antagonistic role throughout the manuscript (e.g. genetic evidence, ChIP for heterochromatin markers, HP1 localization etc.) so it is certainly occurring. But the epistasis of this interaction is rather complex and not very clear to understand at the mechanistic level. The discussion of putative models (as presented in figure 7) is also lacking an explanation for the interactions observed.*

Our Response: Please see our response to the 2nd comment by Reviewer 2. We agree with the reviewer that the antagonistic genetic interactions between *Top3 β* and RISC are not clearly understood,

and it will require extensive new *in vitro* experiments to understand the coordinated actions by Top3 β and RISC.

As indicated in the text, the antagonistic interactions have also been observed between Top3 α and BLM DNA helicase. We have therefore proposed a similar model to explain the antagonistic interactions between Top3 β and RISC, which contains p68 RNA helicase. We agree that this model is imperfect, but it at least provides a working hypothesis that can be tested by people in the field.

To explain why Top3 β mutations can bypass the need of RISC in heterochromatin formation, we hypothesize that there exist other pathways for heterochromatin formation and transcriptional silencing, such as those mediated by piRNA and tRFs (tRNA fragments) machineries. Inactivation of both Top3 β and RISC may trigger activation of an alternative silencing pathway. One such pathway is PIWI-mediated gene silencing, as PIWI has been shown to interact with FMRP to promote heterochromatic gene silencing. We have preliminary data to show that Top3 β -TDRD3 also interacts with PIWI in *Drosophila* germ cells. A future study will test whether Top3 β -TDRD3 also works with PIWI or other pathways in gene silencing.

To address reviewer's comments, we have now added more details in the discussion to help readers understand that activation of the alternative heterochromatin silencing pathways may account for the gene silencing observed in the Top3 β -RISC double mutant. Genetic screens in the Top3 β -RISC double mutant may help to identify this alternative pathway in the future.

4) The finding that Ago binding to Top3 β , unlike other RISC components, is not RNase sensitive (Sup) is quite intriguing. But the authors do not even mention this in the results description. Can the authors comment on that?

Our Response: We have added more explanation to this finding in our revised manuscript. The results suggest a direct protein-protein interaction between Top3 β and AGO2 may exist, which is not mediated or stabilized by RNA. We thank Reviewer 2 for reminding us of this result.

5) The conclusion that "RISC plays more important roles than Top3 β in HP1 recruitment" (page 11) needs additional data to support the claim. Comparative analysis of quantitative value arising from different are rather tricky to evaluate and the authors may wish to consider keeping these comparative approaches as a description note, rather than a stand-alone conclusion. (on a side note, the excessive use of latter/former in the first paragraph makes it very difficult to read...)

Our Response: As requested, we have removed the conclusive statement from the text. We added a more descriptive statement "HP1 recruitment to heterochromatin is more defective in AGO2 than Top3 β mutant". We also replaced several "latter and former" with the real names.

6) The data for gene expression at the sub-telomeric regions is very interesting (figure 7). Although the HP1 binding suggest that these regions are not subjected to the same antagonistic role between Top3 β & RISC (fig.5), it would be interesting to test at the transcript level is any change is detected in the double mutants.

Our Response: We have performed RNA-seq analysis for single and double mutants of *Top3β* and *AGO2*, and found that the levels of the transcripts from the sub-telomeric genes of chromosome 3L and 3R did not show antagonistic interactions. We did not include the data for *AGO2* single and the *Top3β*;*AGO2* double mutant in the paper, because these data were largely negative in nature, and may distract readers from the main point of the paper, which is the HP1 recruitment at pericentric heterochromatin.

7) The authors should state the type of alleles used (null, truncations, point mutations, etc) in the M&M rather than simply referring to the original source.

Our Response: *We revised and mentioned the full genotype of the alleles as you suggested.*

Reviewer #3 (Remarks to the Author):

In the manuscript "A Dual-Activity Topoisomerase Interacts with RNAi Machinery to Promote Heterochromatin Formation and Transcriptional Silencing" by Lee et al., the authors studied the Topoisomerase 3β role in its interaction with RNA and the RISC complex as well as its relevance in transcriptional silencing maintenance.

This work provides important novelties in the field. Beside to confirm in Drosophila the Topoisomerase 3β interaction with TDRD3 already observed in human [Xu et al., Nat Neurosci. 2013; 16(9):1238-47], it shows the interaction with the RISC complex on biochemical basis. In addition, the biochemical and genetic data concerning the transcriptional silencing maintenance involving TDRD3, are important and innovative.

The involvement of Topoisomerase 3β in the epigenetic control of transcription may overcome the specific interest of the field.

The data presented are based on well performed experiments and most of them sound with the conclusions of the manuscript.

However, in order to satisfy the standard quality of Nature Communications, I would suggest some specific modifications and additional experiments:

#1. This work underlines the Topoisomerase 3β role in transcriptional silencing control. From the data presented, it is not evident whether this control derives from the Topoisomerase 3β catalytic activity or from its sole presence (scaffold activity). In order to clarify this point, authors should provide experiments, analogous to those presented in Fig. 3 and 4, studying a Topoisomerase 3β mutant in which the catalytic tyrosine (Ahmad et al., Nucleic Acids Res. 2017 17;45:2704-2713) is replaced with phenylalanine at position 322 (Y332F) to show how Topoisomerase 3β behaves in these controls.

Our Response: We thank the reviewer for his/her encouraging comments and suggestions. As described in our Response to Reviewer 1, we have performed the requested experiments, and found that compared to the wildtype protein, Y332F catalytic mutant of *Top3β* is defective in HP1 recruitment to specific loci in heterochromatin by CHIP-seq assays, and in silencing of two transposable elements. The data indicate that the catalytic activity of *Top3β* is required to promote heterochromatin formation and transcription silencing.

#2. Authors claim that Topoisomerase 3β, together with RISC, promotes HP1 recruitment to pericentric heterochromatin. This is mostly derived from CHIP seq results obtained with antibodies against HP1 in

Topo3β^{-/-} or AGO2^{-/-} mutants. A ChIP-seq using antibodies against Topoisomerase 3β and AGO2 should be performed in order to show a Topoisomerase 3β and RISC presence overlapping to that of HP1. This would more directly show the point.

Our Response: As requested, we have added new ChIP-seq data for Top3β and Flag-tagged AGO2 in *Drosophila* heads. These data reveal several interesting findings:

1. AGO2 and Top3β ChIP islands are present in heterochromatin at much lower frequency and scores than those of HP1 islands (Fig. S6A). They are also present at much lower frequency than the HP1-reduced islands in heterochromatin (Fig. 5G, 5E). The data are consistent with a mechanism of “nucleation and spreading³²”: RISC and Top3β may bind a small number of loci to nucleate the initial assembly of heterochromatin components, which may then recruit additional heterochromatin components to spread to other regions.
2. Almost all AGO2 ChIP islands in pericentric heterochromatin (98%) overlap with those of HP1 (new Fig. 5G), consistent with a role of chromatin-bound RISC in HP1 recruitment to heterochromatin. In contrast, a much smaller percentage of Top3β islands (8%) overlap with those of HP1 (Fig. 5G) or AGO2 (Fig.S6D), arguing that only a minor fraction of chromatin-bound Top3β may be involved in the same process. Inspection of selected regions in heterochromatin revealed that some HP1-reduced peaks in *Top3β* and *AGO2* mutants overlap with AGO2 and Top3β binding sites (Figure 5B-D), supporting the notion that RISC and Top3β work together to promote HP1 recruitment.
3. The majority of HP1 islands (>80%) do not overlap with either AGO2 or Top3β islands (Fig. 5G). The data are largely consistent with the findings that majority of HP1 islands remain unchanged in either mutant (Fig. 5E), suggesting existence of RISC- and Top3β- independent pathways for HP1 recruitment to heterochromatin.
4. Most of Top3β and AGO2 ChIP signals are located in euchromatin, in contrast to those of HP1, which are highly enriched in heterochromatin (New Fig. S6A), suggesting that most of Top3β and AGO2 may function in regulating euchromatic gene expression, but not HP1 recruitment in heterochromatin. Consistent with this notion, only a minor fraction (16%) of AGO2 islands overlap with those of HP1 in euchromatin (Fig. S6B), which is much smaller than the 98% observed in heterochromatin. Our data on AGO2 are consistent with two previous studies showing that AGO2 has functions independent of HP1 recruitment, in transcription, RNA splicing, and chromatin insulation (Moshkovich et al., G&D., 2011; Taliaferro et al., G&D 2013).
5. Our analyses also revealed that a large fraction of Top3β binding sites in euchromatin are present at transcription start sites, suggesting that Top3β may play a role in transcription initiation. We decided not to include these data, because it is not relevant to the theme of this paper. We hope to include these data in a future paper that focuses on the role of Top3β in transcription activation.

Minor modifications required.

Top3β and RISC seem to contribute to the recruitment of HP1 to pericentric and telomeric heterochromatin. However their interaction is antagonistic only for pericentric but not for telomeric heterochromatin: the title of this latter paragraph (pag 12, line 308) should be corrected to emphasize this conclusion.

Our Response: We have changed the subtitle as requested.

Figure S4 shows distribution of H3K9 methylation and HP1 presence in WT and Topo3 β ^{-/-} mutant cells. The reduction of the H3K9me2 and HP1 signals in Topo3 β ^{-/-} mutant should be presented as merged images in order to confirm the effect of Topo3 β ^{-/-} mutation.

Our Response: We were unable to perform the co-staining experiment because the two antibodies are from the same animal host. However, to normalize the signals between wildtype and Topo3 β mutant cells, we included co-staining of histone H3 as an internal control. The results confirm the CHIP-seq data that HP1 and H3K9 methylation signals are reduced in pericentric heterochromatin.

There is a quite extended series of data concerning HP1 sites observed in euchromatin regions. This seems to be a minor effect of the processes described. Actually, authors never mention it in the discussion. To move these data to supplemental informations would be more appropriate.

Our Response: As requested, we have moved the euchromatin data to Supplemental figures.

Finally, we want to thank all three reviewers for the thoughtful suggestions. We believe that the new data based on these suggestions have significantly improved the quality of the manuscript, and hope that it can be accepted for publication now.

Reviewer #1 (Remarks to the Author):

The authors have addressed this reviewer's previous comments and concerns. I have no further requests.

Reviewer #2 (Remarks to the Author):

In this revised version of the manuscript, the authors addressed one of my major points, namely they now provide evidence for how specific mutations in Topoisomerase 3 beta affect the described phenotypes. The experiments were mostly restricted to HP1 localization and TE silencing (and are indeed convincing). It is a shame the authors do not provide evidence regarding the most (PEV). I understand this would require additional strains, but it would be a matter of crosses since all the critical strains are already produced. Perhaps more importantly is the fact that in these new experiments the best evidence for a direct role in RNA topology arises from the deltaRGG mutants. Yet, this mutant is defective in topoisomerase catalytic activity (both towards RNA and DNA). It therefore remains to be addressed whether or not the novel function reported here relate to activity on RNA or DNA. The authors should discuss this openly. Having said so, the data provided is sufficient to support the idea that the catalytic activity of top3beta is required for some of the reported phenotypes (which are indeed novel and highly relevant findings to the field).

The part that remains less clear is indeed the complicated epistasis of the interactions observed. I agree with the authors' reasoning that full understanding of the antagonistic interaction between Top3beta and RISC could be solved in a separate study (and I also agree this one is already very extended). But in this case I would strongly advice to tone down the discussion on the findings related to the double mutants. They appear presented with a conclusive tone (including explicit description in the abstract). Yet, as the authors admit, the exact reason for this complicated interaction remains to be addressed. This should also be properly acknowledged in the text.

Reviewer #3 (Remarks to the Author):

The revised version of the manuscript "Topoisomerase 3 β interacts with the RNAi mechanism to promote the formation of heterochromatin and transcriptional silencing" satisfies all the specific

points raised compared to the previous version. The results of new experiments are consistent with the general structure of the work and there are no contradictions. According to these considerations, in my opinion, the work can be considered for publication.

Point-by-point response to Reviewer 2's comments:

Reviewer 2: In this revised version of the manuscript, the authors addressed one of my major points, namely they now provide evidence for how specific mutations in Topoisomerase 3 beta affect the described phenotypes. The experiments were mostly restricted to HP1 localization and TE silencing (and are indeed convincing). It is a shame the authors do not provide evidence regarding the most (PEV). I understand this would require additional strains, but it would be a matter of crosses since all the critical strains are already produced. Perhaps more importantly is the fact that in these new experiments the best evidence for a direct role in RNA topology arises from the deltaRGG mutants. Yet, this mutant is defective in topoisomerase catalytic activity (both towards RNA and DNA). It therefore remains to be addressed whether or not the novel function reported here relate to activity on RNA or DNA. The authors should discuss this openly. Having said so, the data provided is sufficient to support the idea that the catalytic activity of top3beta is required for some of the reported phenotypes (which are indeed novel and highly relevant findings to the field).

Our response: We thank reviewer 2 for his positive comments. Per his request, we have added a statement in the Discussion indicating that “because the RGG-box deletion mutant not only has defective RNA-binding activity, but also reduced catalytic activity for both DNA and RNA, it is possible that the reduced catalytic activity on either DNA, or RNA, or both, is responsible for the defective heterochromatin formation. A separation-of-function mutant, which inactivates the catalytic activity of Top3b on one nucleic acid but not the other, is needed to clarify this issue.”

Reviewer 2: The part that remains less clear is indeed the complicated epistasis of the interactions observed. I agree with the authors' reasoning that full understanding of the antagonistic interaction between Top3beta and RISC could be solved in a separate study (and I also agree this one is already very extended). But in this case I would strongly advice to tone down the discussion on the findings related to the double mutants. They appear presented with a conclusive tone (including explicit description in the abstract). Yet, as the authors admit, the exact reason for this complicated interaction remains to be addressed. This should also be properly acknowledged in the text.

Our Response: Per reviewer's request, we have toned down the conclusion regarding the double mutants and the antagonistic genetic interactions between Top3b and RISC throughout the manuscript. First, we have removed the statement regarding the double mutant in the Abstract. In the last sentence of the Abstract, we have also toned down the conclusion. Now it reads “Our data suggest that Top3 β may act as an RNA topoisomerase in siRNA-guided heterochromatin formation and transcriptional silencing.” Second, throughout the text, we have replaced “antagonistically interact” with “genetically interact”. This change should de-emphasize the antagonistic nature of the genetic interactions observed in the double mutant, while still illustrate that the two genes genetically interact. Third, we have re-written the paragraph in Discussion regarding the Top3b-p68 double mutant. We have shortened our discussion on our model on how two proteins may work together. We also added the statement indicating that our hypothesis does not explain well why the p68 and Top3b double mutant exhibits suppression of the defective heterochromatin observed in the single mutant. We stated clearly what we proposed is a hypothesis, which needs to be verified by future experiments.

Finally, we greatly appreciate the constructive comments and very thoughtful suggestions by reviewer 2.